# *Ustilago maydis* PR-1-like protein has evolved two distinct domains for dual virulence activities

Yu-Han Lin [1,3], Meng-Yun Xu[1,3], Chuan-Chih Hsu [1,3], Florensia Ariani Damei [1], Hui-Chun Lee[1], Wei-Lun Tsai[1], Cuong V. Hoang [1], Yin-Ru Chiang[2] & Lay-Sun Ma [1] ✉

The diversification of effector function, driven by a co-evolutionary arms race, enables pathogens to establish compatible interactions with hosts. Structurally conserved plant pathogenesis-related PR-1 and PR-1-like (PR-1L) proteins are involved in plant defense and fungal virulence, respectively. It is unclear how fungal PR-1L counters plant defense. Here, we show that *Ustilago maydis* UmPR-1La and yeast ScPRY1, with conserved phenolic resistance functions, are Ser/Thr-rich region mediated cell-surface localization proteins. However, UmPR-1La has gained specialized activity in sensing phenolics and eliciting hyphal-like formation to guide fungal growth in plants. Additionally, *U. maydis* hijacks maize cathepsin B-like 3 (CatB3) to release functional CAPE-like peptides by cleaving UmPR-1La's conserved CNYD motif, subverting plant CAPE-primed immunity and promoting fungal virulence. Surprisingly, CatB3 avoids cleavage of plant PR-1s, despite the presence of the same conserved CNYD motif. Our work highlights that UmPR-1La has acquired additional dual roles to suppress plant defense and sustain the infection process of fungal pathogens.

Proteins containing CAP (Cysteine-rich secretory protein/Antigen 5/Pathogenesis-related 1) domains are present in all kingdoms of life. They regulate diverse biological processes, including reproduction, immune defense, venom toxicity, fungal virulence, and cancer development[1]. The structurally conserved CAP domain adopts a unique α-β-α sandwich fold stabilized by disulfide bonds and binds lipids[2]. But, how the lipid-binding CAP proteins influence such a wide range of biological processes remains enigmatic.

Pathogenesis-related 1 (PR-1) was initially identified in plants, and homologous PR-1-like (PR-1L) proteins are widespread in a variety of eukaryotes. In plants, the *PR-1* gene is highly induced by salicylic acid (SA) and is a marker for SA-mediated defense responses[3]. PR-1 proteins localize to the apoplast or vacuoles to regulate abiotic and biotic stresses[4,5]. Upon pathogen attack, PR-1 proteins accumulate in the apoplast to sequester sterols from oomycetes that depend on plant-produced sterols for survival, leading to growth inhibition[6,7]. Evidence

suggests that PR-1 harbors a conserved CNYx motif in the CAP domain, which can be cleaved to release an 11-amino acid peptide[8-10]. These CAP-derived peptides (CAPE; PxGNxxxxxPY) enhance plant immunity[8,9] or negatively regulate salt-stress tolerance[11]. However, it is unclear whether a protease is involved in the cleavage of the CNYx motif and if there is a receptor to perceive CAPE peptides that activate plant immunity.

In yeast, the CAP domain of PRYs (Pathogen-Related Yeast) consists of two distinct sites – a caveolin-binding motif (CBM) and a hydrophobic pocket for binding to sterols and fatty acids, respectively[12,13]. PRY1/2/3 proteins are capable of exporting sterols and detoxifying small hydrophobic compounds[14,15]. In addition, PRY3 has also been reported to inhibit yeast mating[15]. While most research has focused on plant PR-1 and yeast PRYs, less attention has been paid to fungal PR-1L proteins. In contrast to the anti-pathogen function of PR-1, fungal PR-1L proteins from *Fusarium oxysporum*, *Moniliophthora*

[1]Institute of Plant and Microbial Biology, Academia Sinica, Taipei 115201, Taiwan. [2]Biodiversity Research Center, Academia Sinica, Taipei 115201, Taiwan. [3]These authors contributed equally: Yu-Han Lin, Meng-Yun Xu, Chuan-Chih Hsu. ✉e-mail: laysunma@gate.sinica.edu.tw

*perniciosa*, and *Candida albicans* have emerged as novel lipid-binding virulence factors that are highly expressed during infection[16–20]. Despite contributing to fungal virulence, their action mechanisms remain unknown.

The corn smut *Ustilago maydis* is a dimorphic fungus switching from a unicellular haploid cell to pathogenic dikaryotic hyphae when infecting maize[21]. This irreversible morphological shift is essential for *U. maydis* virulence. During the biotrophic stages of its sexual life cycle, *U. maydis* requires fine sensing of environmental signals to adapt to changes in plant development and direct hyphal growth toward nutrient sources[22]. It sequentially secretes a pool of effector proteins to shield hyphae from plant attacks, suppress immune responses, and manipulate plant metabolism, ultimately facilitating successful colonization of host plants[23]. Effectors that overcome plant resistance are usually considered the most rapidly evolved genes in coevolutionary arms races. However, it is unknown whether these effector proteins contribute to the integration of plant signals to promote *U. maydis* virulence.

In this study, we functionally characterize the secreted effector protein UmPR-1La from *U. maydis* to understand how the structurally conserved CAP domain of a PR-1-like protein evolved an opposite role to plant PR-1 proteins. We demonstrate that UmPR-1La promotes virulence through dual functional activities derived from two distinct motifs. UmPR-1La binds plant-derived phenolics to elicit hyphal-like structures, while a CAPE-like peptide, released from UmPR-1La via the action of a plant cysteine protease, can suppress plant immunity. Our work provides a mechanistic understanding of the fungal parasitism enabled by the dual roles of PR-1-like protein.

## Results

### *U. maydis* PR-1-like protein is a virulence factor

*Ustilago maydis* encodes two CAP-domain containing PR-1-like proteins, designated UmPR-1La (UMAG_01204) and UmPR-1Lb (UMAG_04343). They consist of an N-terminal signal peptide, a serine/threonine (Ser/Thr)-rich extension region, and a C-terminal CAP domain (Fig. 1a). This extension region is also present in three out of the 17 maize PR-1 proteins[24]. The CAP domain sequence alignment showed that UmPR-1La, but not UmPR-1Lb, contains a conserved CNYx motif also found in plant PR-1s (Supplementary Fig. 1a, Fig. 1a). This suggests that a 16-amino acid peptide (PPGNYIGKFKENVSPN) following the motif may be released from UmPR-1La by a protease. Comparing the peptide sequences following the CNYx motif in plant PR-1 and smut fungal PR-1L proteins, a notable conservation was observed in the first seven amino acids (P[P/V]GN[Φ/V][I/V]G) (Fig. 1b). We designate the fungal peptides as CAPE-like peptides (CAPE-L). To understand the evolutionary relationship between smut fungal PR-1Ls versus *Saccharomyces cerevisiae* PRYs and plant PR-1s, we performed a phylogenetic analysis using CAP domains from several plants and fungal species (Fig. 1c). UmPR-1La and UmPR-1Lb are placed in two separate clades (clades I and II) of smut fungal PR-1L proteins. Both of these clades are distantly related to the plant PR-1 clade III. Clade I, which includes UmPR-1La, is more closely related to the ScPRY clade IV than to the smut fungal clade II. The analysis of the CAP domains infers protein evolution, which shows that the UmPR-1La and UmPR-1Lb proteins belong to two subfamilies of PR-1L proteins expanded in smut fungal pathogens. The expansion and sequence divergence of smut fungal PR-1L proteins suggest that UmPR-1La and UmPR-1Lb may have distinct functional roles.

A qRT-PCR analysis showed that *PR-1La* and *PR-1Lb* were not expressed in axenic culture, but were induced during biotrophic development of *U. maydis* SG200, a solopathogenic haploid strain that completes its life cycle without a mating partner[25] (Fig. 1d and Supplementary Fig. 1b). The deletion of *PR-1La*, but not *PR-1Lb*, in SG200 resulted in a decrease in disease symptoms in infected plants (Fig. 1e and Supplementary Fig. 1c). This defect could be rescued by

introducing a single copy of *PR-1La* into the Δ*pr-1la* mutant under the control of its native promoter. Moreover, a similar decrease in virulence was also observed in the double mutant ΔΔ*pr-1lab*. While grown in a liquid medium or induced to form filaments on charcoal agar, the deletion mutants and complemented strains were indistinguishable from SG200 (Supplementary Fig. 1d). These findings suggest UmPR-1La is associated with the *in-planta* development of *U. maydis*.

### The Ser/Thr-rich region mediates cell surface exposure of CAP domain

Glycosylphosphatidylinositol (GPI)-anchored ScPRY3 and *C. albicans* non-GPI-anchored Rbe1p are cell surface-localized PR-1L proteins[15,26]. It has also been reported that Ser/Thr-rich regions of GPI-anchored glycoproteins could override the role of GPI anchors in surface exposure in a length-dependent manner[27]. We, therefore, hypothesized that the Ser/Thr-rich region of non-GPI anchored UmPR-1L and ScPRY1 proteins localizes CAP domains to the cell surface. As the expression of *U. maydis PR-1L* genes was not detected in axenic culture, we constitutively expressed both UmPR-1L proteins with C-terminal-HA-tags in SG200 under the control of the *otef* promoter. For a comparative assessment of localization by fluorescence microscopy, an ScPRY1-mCherry fusion protein was also constitutively expressed in yeast AH109 cells using the *ADH1* promoter. Both UmPR-1La and ScPRY1-mCherry localized to the surface of SG200 filaments and AH109 cells, respectively (Fig. 2a, b). To determine the localization of UmPR-1La on filamentous cell walls after plasmolysis, we utilized the haploid AB33 strain. This strain harbors the compatible *bW1/bE2* genes controlled by the nitrate-inducible *nar1* promoter, enabling filamentous growth in a liquid nitrate minimal medium[28]. After subjecting the AB33_PR-1La filaments (overexpressing HA-tagged PR-1La) to plasmolysis, the AF594-immunostained PR-1La proteins were not located inside the plasmolysis-expanded region (Fig. 2c). Instead, they showed co-localization with wheat germ agglutinin (WGA)-AF488 staining chitins on the surface of the region, confirming the presence of UmPR-1La on the cell walls.

In contrast, UmPR-1Lb proteins were not found on the cell surface due to their instability. Only truncated fragments were detected after secretion (Fig. 2a, d). Interestingly, when ScPRY1 was fused with the signal-peptide (SP) of UmPR-1La and constitutively expressed in *U. maydis*, it did not localize to the cell surface, despite being expressed and secreted (Fig. 2a, d). Possibly the PRY1 ligand required for cell surface binding is absent in *U. maydis* or inaccessible to ScPRY1. The cell-surface localization of ScPRY1 in *U. maydis* was restored by fusing the N-terminus of UmPR-1La containing the SP and Ser/Thr-rich region to the ScPRY1 CAP-domain ($N_{PR-1La}$-$CAP_{PRY1}$) (Fig. 2a). The analysis indicated that the Ser/Thr-region is sufficient to mediate the cell surface attachment of UmPR-1La in *U. maydis*.

To further investigate how PR-1La binds to the surface of *U. maydis* cells, we purified each recombinant protein PR-1La (PR-1La24–279; FL), the Ser/Thr region (PR-1La24–153; S/T), and the CAP-domain (PR-1La126–279; CAP) and incubated with SG200 filaments. Only FL and S/T proteins were bound to the cell surface (Fig. 2e). This finding further supports the notion that the N-terminal Ser/Thr-extension region is necessary and sufficient to mediate the cell wall localization of PR-1La. Interestingly, the localization of UmPR-1La in the bud necks and growing tips of *U. maydis* sporidial cells coincided with sites of chitin and chitosan deposition[29] (Fig. 2a). This suggests that UmPR-1La proteins may associate with fungal cell walls via binding to chitin/chitosan. Alternatively, it may interact with the cell wall proteins.

### UmPR-1La protects hyphae against plant toxic phenolics

We next investigated the biological relevance of the CAP domain at the fungal cell wall. ScPRY1 and ScPRY2 protect cells against eugenols in a caveolin-binding motif (CBM)-dependent manner[12,14], and UmPR-1La was relatively close to ScPRYs in our phylogenetic analysis (Fig. 1c).

Therefore, we hypothesized that UmPR-1La could shield cells from toxic plant compounds. Based on sequence alignment of the CBMs from known lipid-binding PR-1L proteins[19], we identified three aromatic residues at positions 1, 3, and 8 of the CBM, which are crucial for sterol/phenolic binding in ScPRY1[12] and are conserved in UmPR-1La (Fig. 3a). In contrast, UmPR-1Lb does not exhibit conservations at positions 3 and 8. The analysis suggests that UmPR-1La may bind to sterols or phenolics, potentially protecting *U. maydis* cells from their toxicity. To test this idea, we treated SG200 cells expressing PR-1L proteins with eugenols, which maize also produces[30]. Only cells expressing UmPR-1La proteins induced pseudohyphal structures and survived (Fig. 3b). In contrast, cells producing UmPR-1Lb, ScPRY1, or chimeric proteins ($N_{PR-1La}$-$CAP_{PRY1}$) did not form pseudohyphae and lost the ability to grow after eugenol treatment.

These observations led us to investigate whether *S. cerevisiae* cells could also induce pseudohyphae in response to eugenols. Despite surviving the eugenol treatment, the *S. cerevisiae* cells were unable to develop pseudohyphal structures, even when overexpressing ScPRY1-mCherry (Supplementary Fig. 3). The result suggests that UmPR-1La possesses an additional specialized function to induce pseudohyphal structures in *U. maydis*, which is absent in ScPRY1, and this structure is not needed to protect *S. cerevisiae* cells. This finding also explains why the PRY1 and chimeric proteins, which contain the PRY1 CAP domains capable of binding eugenols[14], fail to shield *U. maydis* from eugenols, despite the conservation of the protective role of PR-1-like proteins between *U. maydis* and *S. cerevisiae*.

We further mutated the aromatic residues at positions 3 and 8 of the UmPR-1La CBM to alanine (PR-1La$^{F212A Y217A}$; FY*). UmPR-1La (FY*) was

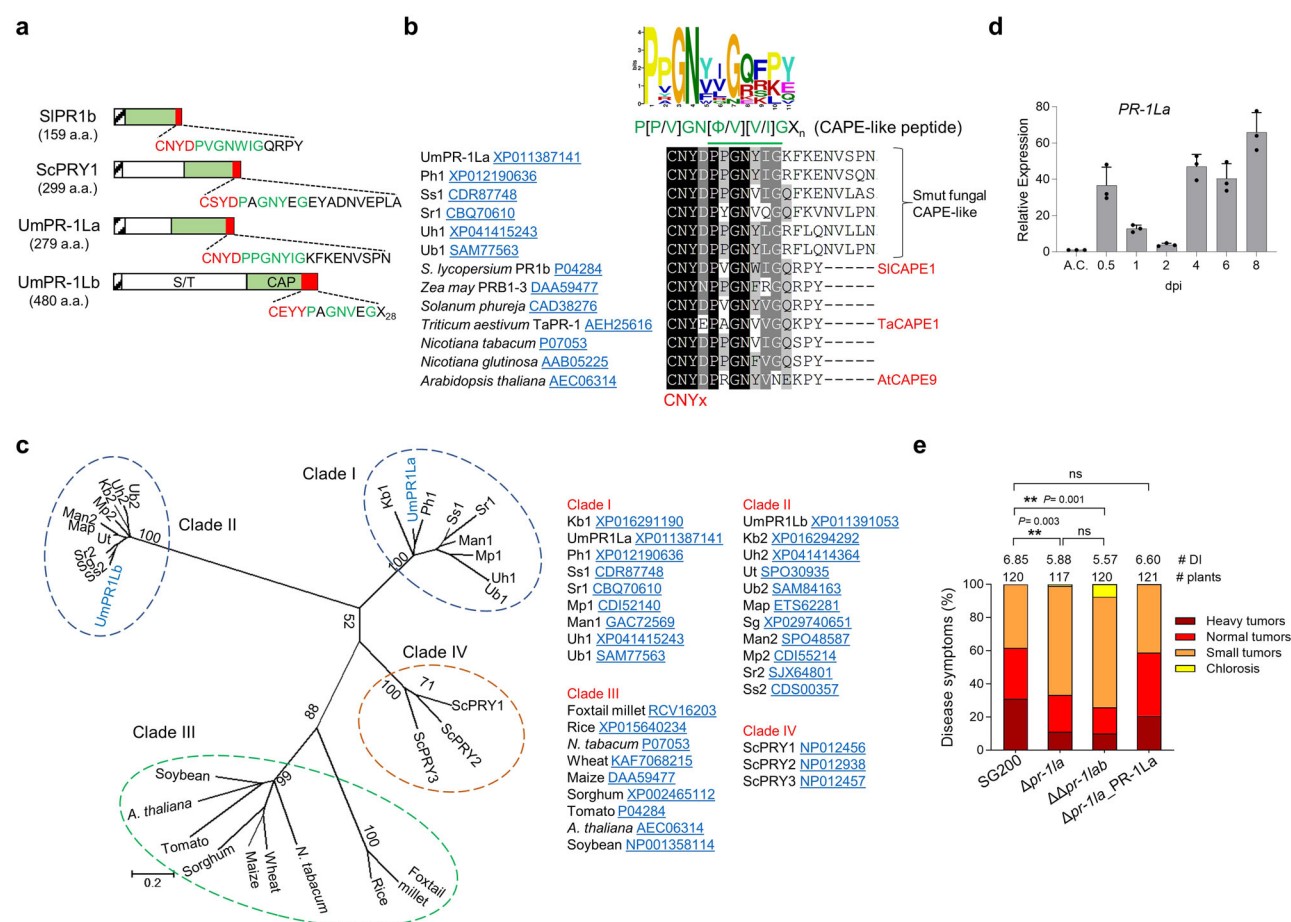

**Fig. 1 | The PR-1-like (PR-1L) family proteins in *U. maydis*. a** Schematic drawing displays plant PR-1 and PR-1L protein domain structure. The signal peptide (SP) is depicted as a black stripe, the Ser/Thr-rich (S/T) region as a white box, and the CNYx motif and CAPE/CAPE-like peptide (a red box) are located at the C-terminus of the CAP-domain (green box). Conserved residues shared between plant PR-1's CAPE and PR-1L's CAPE-like peptide are marked in green. *Solanum lycopersium* SlPR1b (#P04284); *Saccharomyces cerevisiae* ScPRY1 (#NP012456). **b** The sequence comparison of plant CAPEs and smut fungal CAPE-like peptides after CNYx motif. The green line indicates the MEME-derived consensus sequence P[P/V]GN[Φ/V][V/I]G. The protein accession numbers are provided. Plant CAPE peptides that have been identified and reported are indicated[8,9]. Ph *Pseudozyma hubeiensis*, Ss *Sporisorium scitamineum*, Sr *Sporisorium reilianum*, Uh *Ustilago hordei*, Ub *Ustilago bromivora*. **c** Phylogenetic analysis of plant PR-1 and PR-1-like proteins from smut fungi and yeast. Protein sequences are retrieved from the NCBI, aligned using Clustal Omega, and curated with trimAl. Analysis was carried out in MEGA 7.0 using the Maximum Likelihood method with the WAG+G+I model and 1000 bootstrap replicates. Clade I and II: smut fungal PR-1Ls; Clade III: Plant PR-1s; IV: yeast. Kb *Kalmanozyma brasiliensis*, Mp *Melanopsichium pennsylvanicum*, Man *Moesziomyces antarcticus*, Map *Moesziomyces aphidis*, Ut *Ustilago trichophora*, Sg *Sporisorium graminicola*. **d** qRT-PCR analysis of *PR-1La* expression. Total RNA was extracted from SG200-infected maize leaves harvested at the indicated days post-infection (dpi), and from cells cultured in liquid culture (A.C.). *Peptidylprolyl Isomerase (PPI)* gene was used for normalization. The *PR-1La* expression level in A.C. was set to 1.0. Values are mean ± standard deviation (SD) of three biological replicates. **e** Virulence assay of Δ*pr-1l* and the complementation strain. Δ*pr1la*_PR-1La: Δ*pr1la* was complemented by introducing a single allele of *PR-1La* under the control of its native promoter. Disease symptoms were scored at 12 dpi according to the severity depicted in the color code. The number of infected plants and the average disease index (DI) from four independent infections are shown. Significant differences between samples were determined by a two-tailed unpaired t-test (*P < 0.05; **P < 0.01; ***P < 0.001). ns no significance. Source data are provided as a Source Data file.

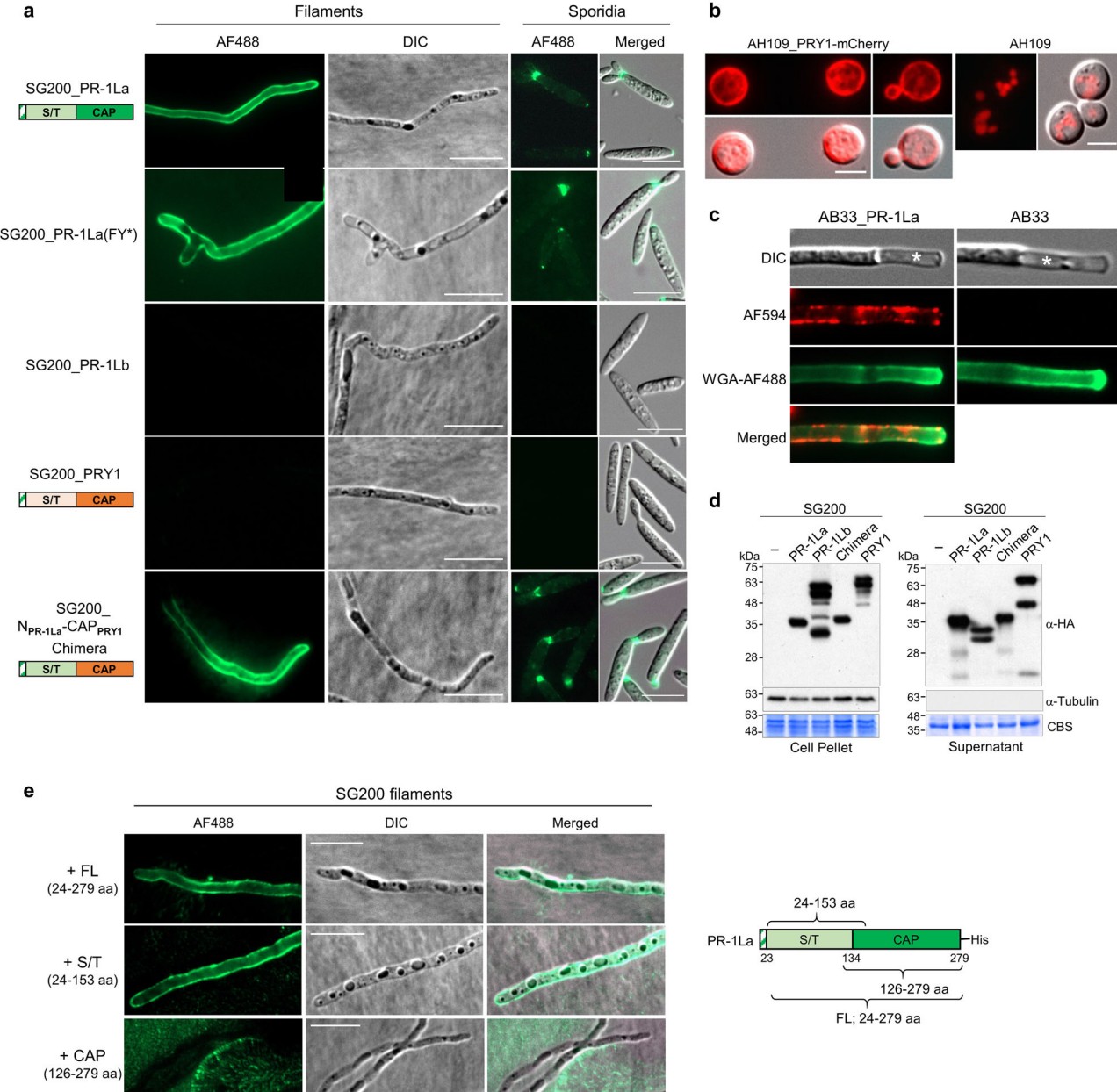

**Fig. 2 | UmPR-1La is a cell-surface-associated protein. a** UmPR-1La localizes to the surface of filaments and the bud-neck of sporidial cells. Hydroxyl fatty acid-induced SG200 expressing C-terminal HA-tagged proteins under the *otef* promoter formed filaments on a hydrophobic parafilm surface. Immunolocalization was performed using anti-HA and anti-IgG-AF488 antibodies to locate HA-tagged proteins. SG200_PR-1La: produced the signal-peptide (SP)-containing full-length protein (279 aa; 31.5 kDa); SG200_PR-1La(FY*): produced PR-1La(FY*) proteins with the two aromatic residues at positions 3 and 8 of the caveolin binding motif replaced with alanine (see Fig. 3a); SG200_PR-1Lb: produced the SP-containing full-length protein (480 aa, 54.2 kDa); SG200_PRY1: produced the proteins (32.2 kDa) containing UmPR-1La's SP (green strip box) fused to PRY1 (20-299 aa); SG200_N$_{PR-1La}$-CAP$_{PRY1}$: produced the chimera proteins (31 kDa) containing the N-terminus of UmPR1-La (1–133 aa, including SP and S/T region (green box)) fused to the PRY1's CAP-domain (159–299 aa; orange box). Bars, 10 μm. aa: amino acid. **b** PRY1-mCherry localizes to the periphery of AH109 cells. Cells expressing PRY1-mCherry proteins driven by the *ADH1* promoter were cultured on YPD agar and subjected to fluorescence detection. Bars: 10 μm. **c** UmPR-1La colocalizes with chitins. AB33 (negative control) and AB33_PR-1La strains expressing C-terminal HA-tagged PR-1La under the *otef* promoter, were grown in nitrate minimal medium to induce filaments[71]. Immunolocalization of PR-1La was performed using anti-HA and anti-IgG-AF594, followed by chitin staining with WGA-AF488. Plasmolysis was performed before fluorescence detection. Plasmolysis-enlarged regions (*). **d** Secretion of PR-1L, PRY1, and chimeric proteins. SG200 expressing HA-tagged proteins under the *otef* promoter were grown in YEPSL liquid medium until reaching an OD$_{600}$ of 0.6. Proteins from cell pellets and TCA-precipitated supernatants were prepared for immunoblotting analysis. Tubulin served as a cytosolic protein control, and the Coomassie blue staining (CBS)-membranes were used to verify loading. **e** PR-1La binds to the filamentous surface via its N-terminus. Immunolocalization of PR-1LaHis on SG200-filaments using anti-His and anti-IgG-AF488 antibodies. Bars, 10 μm. A diagram illustrates the positions of full-length and truncated PR-1La proteins. The protein purity is shown in Supplementary Fig. 2a. All Fig. 2 experiments were performed independently at least three times, yielding consistent outcomes. Source data are provided as a Source Data file.

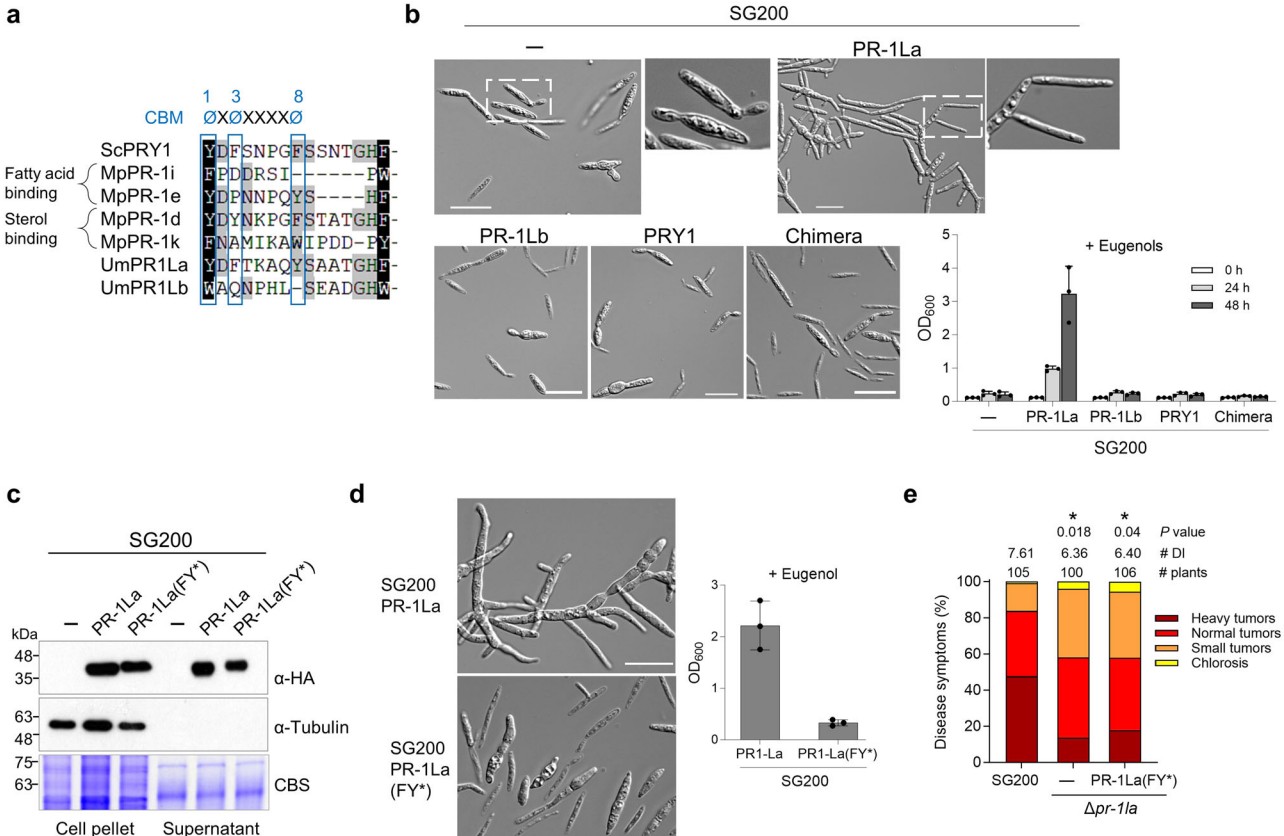

**Fig. 3 | UmPR-1La's CBM is essential for conferring resistance against eugenol.**
**a** Sequence alignment analysis of the CBM of fungal PR-1-like proteins. *Mon-iliophthora perniciosa* MpPR-1i (AEZ63368) and MpPR-1e (AEZ63364) are fatty-acid binding proteins, while MpPR-1d (AEZ63363), MpPR-1k (AEZ63370) and yeast PRY1 are sterol-binding proteins[14,19]. The positions of aromatic residues (ø) within the CBM are indicated at the top. **b, d** UmPR-1La protects cells against eugenol toxicity. After a 24-h treatment with 1 mM eugenol, cell morphology images of SG200 and SG200 expressing C-terminal HA-tagged proteins under the *otef* promoter were captured. The optical densities ($OD_{600}$) of eugenol-treated cells were assessed at indicated time points (**b**) or 48 h (**d**). Values indicate mean ± sd from three inde-pendent biological assays. Bars, 20 μm. **c** Secretion of PR-1La and PR-1La(FY*). Proteins from cell pellets and TCA-precipitated supernatants of SG200 cells

expressing HA-tagged proteins were prepared for immunoblotting analysis. Tubulin was used as an internal control for a non-secreted cytosolic protein (1:8,000 dilution, Sigma #T6199). Coomassie-blue staining of membrane served as a means of verifying loading. Similar results were observed in two independent biological experiments. **e** Virulence assay of SG200, Δ*pr-1la*, and the com-plementation strain Δ*pr-1la*_PR-1La(FY*). Δ*pr-1la*_PR-1La(FY*): Δ*pr-1la* was com-plemented by introducing a single allele of *PR-1La(FY*)* under the control of its native promoter. Disease symptoms of infected plants were evaluated at 12 dpi. The number of infected plants from three independent infections is shown above each column. The average DI is reported, and asterisks indicate statistical significance compared to SG200 as determined by a two-tailed unpaired Student's *t* test ($p < 0.05$). Source data are provided as a Source Data file.

expressed and secreted in SG200, showing the similar localization pattern as unmutated UmPR-1La (Figs. 2a and 3c). However, it lost the ability to induce pseudohyphal structures to protect SG200 cells against eugenols and rescue the virulence of Δ*pr-1la* (Fig. 3d–e). These results indicate that UmPR-1La promotes fungal virulence by initiating protective hyphal-like structures to safeguard cells against toxic compounds in a CBM-dependent manner. Collectively, these findings also imply that the activation of protective UmPR-1La-shielded hyphal-like structures necessitates cell-wall localization, phenolic binding, and signal transduction. Failure to complete any of these steps will hinder the formation of protective structures crucial for *U. maydis* virulence.

## UmPR-1La senses plant phenolics and elicits pseudohyphae formation

The formation of pseudohyphae in UmPR-1La-expressing cells sug-gests that UmPR-1La plays a role in sensing plant signals to activate the transition of nonpathogenic to pathogenic growth, which is crucial for *U. maydis* virulence. In maize, G-type lignin biosynthesis initiates with ferulic acids derived from the shikimate phenylpropanoid pathway. These ferulic acids undergo several intermediate conversions before ultimately producing lipophilic eugenols[30,31] (Fig. 4a). The discovery of ScPRY1's binding to various structurally related steroids[32] has led us to

speculate that UmPR-1La might respond to eugenol precursors exhi-biting different levels of antifungal activity[33]. SG200_PR-1La cells responded to ferulic acid (FA), coumaric acid (CouA), and coniferyl alcohol (CA) by forming long and branching pseudohyphal structures, similar to the morphology observed in eugenol-treated cells (Fig. 4b and Supplementary Fig. 4a). In contrast, UmPR-1La(FY*) cells showed no induction of pseudohyphae upon exposure to the tested com-pounds. These cells were insensitive to FA and CouA, continuing to produce healthy sporidial cells and increasing in cell density (Fig. 4b and Supplementary Fig. 4a). However, UmPR-1La(FY*) cells treated with CA displayed an unhealthy appearance with swollen and abnor-mal morphology (Supplementary Fig. 4a). Despite these changes, CA-treated UmPR-1La(FY*) cells exhibited a higher survival rate compared to cells treated with eugenol. The phenolic-treatment analysis indi-cates that carboxylic phenolics, such as CouA and FA, are less toxic to *U. maydis*. Despite the varying levels of toxicity, *U. maydis* responds to structurally related phenolics and triggers pseudohyphae via UmPR-1La's CBM.

We next investigated whether Sr10279, an ortholog of UmPR-1La from smut fungus *Sporisorium reilianum* containing both highly con-served CBM and CNYx motifs (Supplementary Fig. 1a), could stimulate hyphal-like formation in *U. maydis* and restore the Δ*pr-1la* virulence.

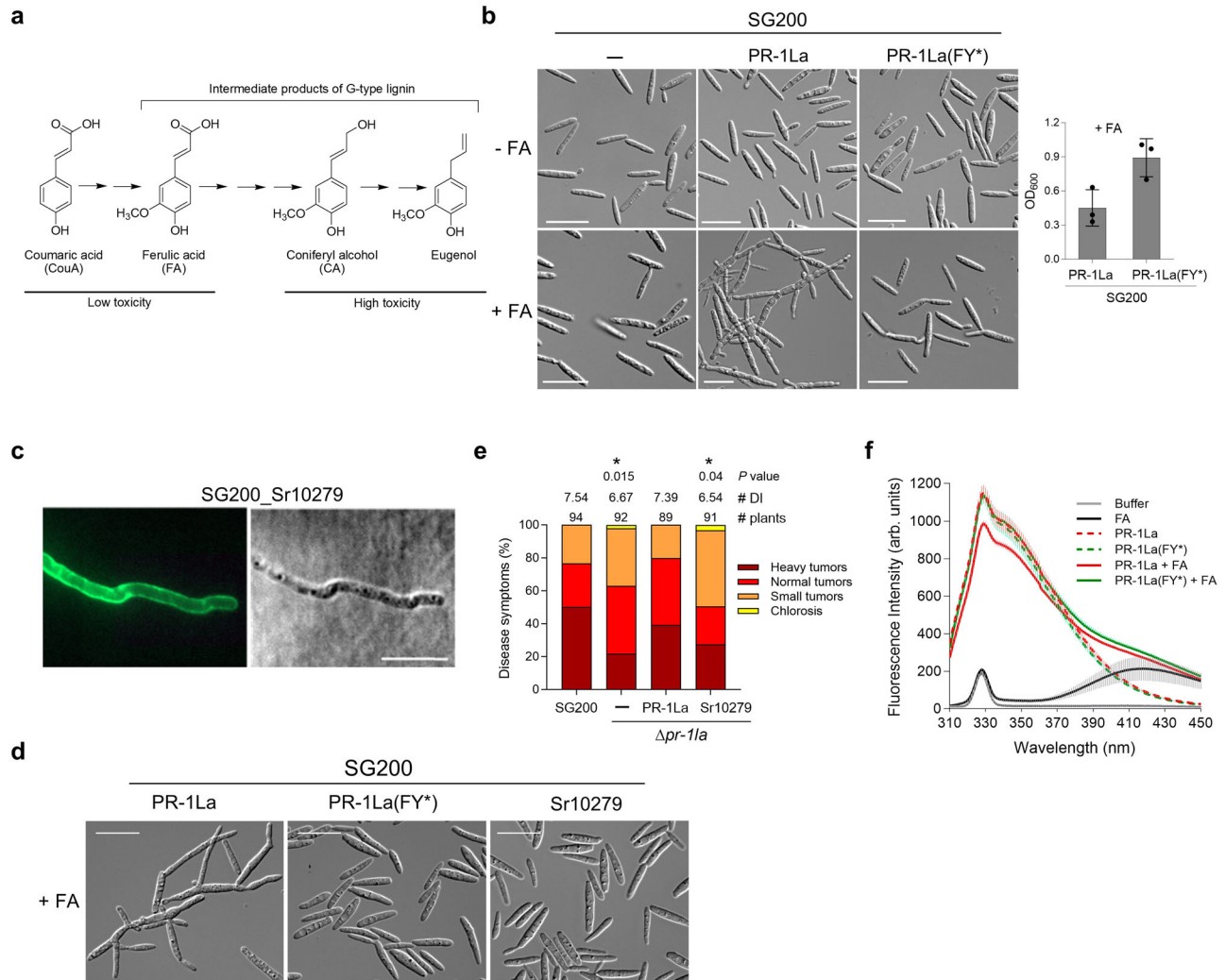

**Fig. 4 | Phenolic binding of UmPR-1La triggers pseudohyphae formation.**
**a** Structures of intermediate phenolic compounds produced in the G-type lignin pathway. The phenolic's toxicity level is determined by the sensitivity of *U. maydis* to the compound. **b** UmPR-1La triggers pseudohyphae formation in response to other phenolics. SG200 and its derivatives expressing the indicated HA-tagged proteins under the *otef* promoter were grown in YEPSL medium with or without 7 mM ferulic acid (FA) for 24 h. Cell morphology images of the indicated strains were captured, and the $OD_{600}$ were measured after 24 h. Values indicate mean ± sd from three independent biological assays. Bars, 20 μm. **c** Sr10279 localizes to the filamentous surface. SG200_Sr10279: the UmPR-1La's SP was fused to C-terminal HA-tagged Sr10279 (accession# CBQ70610) and expressed under the *otef* promoter in SG200. Immunolocalization of Sr10279 on SG200_Sr10279 filaments was performed using an anti-HA and anti-IgG-AF488 antibodies. Bars, 10 μm. **d** Sr10279 fails to trigger pseudohyphae formation. The morphology of cells expressing the indicated proteins, grown in FA-supplemented YEPSL medium for 24 h at 28 °C, was captured. Bars, 20 μm. Similar results were observed in three independent

biological experiments (**c**, **d**). **e** *S. reilianum* Sr10279 fails to restore Δ*pr-1la* virulence. Δ*pr-1la*_Sr10279: a single allele of *Sr10279* containing the *UmPR-1La*'s SP and under the control of *UmPR-1La* promoter was introduced into Δ*pr-1la*. The number of infected plants from three independent infections is indicated above each column. The DI is reported, and asterisks indicate statistical significance compared to SG200 as determined by a two-tailed unpaired *t*-test ($p < 0.05$). **f** Detection of PR-1La binding to ferulic acids. The fluorescence intensities of recombinant PR-1LaHis proteins were measured after incubation with or without 5 μM FA in the sodium acetate buffer (pH 5.5) for 15 min. Buffer control: sodium acetate buffer. FA control: the FA containing buffer. Tryptophan residues in PR-1La proteins were selectively excited at 290 nm, and the emission spectra were recorded in the 310–450 nm range. Values represent mean ± sd from three independent measurements in arbitrary units (arb. units). These measurements were conducted using two separate preparations of purified proteins. Source data are provided as a Source Data file.

The C-terminal HA-tagged Sr10279 proteins expressed under the *otef* promoter localized to the filamentous surface of SG200_SP_{PR-1La}-Sr10279; however, when exposed to FA, they did not form hyphal-like structures (Fig. 4c, d). Furthermore, introducing a single copy of the *Sr10279* allele, regulated by *UmPR-1La*'s promoter, failed to rescue the virulence phenotype of Δ*pr-1la* (Fig. 4e). This discrepancy can be attributed to the potential divergence in downstream signal reception partners between Sr10279 and UmPR-1La, despite their CAP domains likely having highly conserved function. The partner of UmPR-1La is likely unable to interact with Sr10279 and PRY1, leading to their failure in inducing hyphal-like structures in *U. maydis*. While Sr10279 is

incapable of substituting UmPR-1La's role in eliciting hyphal-like structures in *U. maydis*, it may be capable of inducing hyphal-like structures in *S. reilianum* cells.

To investigate the direct binding of PR-1La to phenolics, we performed fluorescence spectroscopy to measure the change in intrinsic tryptophan fluorescence of recombinant PR-1La and PR-1La(FY*) proteins upon binding to FA and CA. Following incubation with fixed or various concentrations of FA, the intrinsic fluorescence intensity of PR-1La decreased in an FA-concentration-dependent manner, while it remained unchanged for PR-1La(FY*) proteins (Fig. 4f and Supplementary Fig. 4b). In the case of CA, both versions of the PR-1La proteins

responded to CA by causing a shift in the fluorescence peak. However, the peak intensity of PR-1La(FY*) was slightly reduced relative to PR-1La (Supplementary Fig. 4c). Notably, the reduction in fluorescence was not due to protein instability during the incubation (Supplementary Fig. 4d). Our finding is in line with the previous report showing that mutations in the CBM of PRY1 affect binding to cholesteryl acetate but not cholesterol[12]. Similarly, our result suggests that *U. maydis* PR-1La directly binds to ferulic acids but may have different binding affinity among the related compounds.

### Cathepsin-B-like protease releases CAPE-La peptides from UmPR-1La

The projection of CAP domains at the cell surface, which fulfills a dual function in detecting phenolics and providing protection, has the potential to expose the CNYx motif to an unidentified protease. This exposure could facilitate the release of UmCAPE-La (PPGNYIGKF-KENVSPN) peptides into the plant apoplast and contributes to *U. maydis* virulence. To investigate the cleavage of UmPR-1La, we collected the apoplastic fluid from SG200-infected leaves, which was subjected to LC-MS/MS analysis to identify the peptides. Despite multiple attempts, we failed to detect endogenous UmCAPE-La peptide, possibly due to the low expression of *PR-1La*. As an alternative approach, we explored UmPR-1La cleavage by incubating the culture supernatant of SG200_PR-1La strain with the apoplastic fluid from salicylic-acid (SA)-inoculated maize leaves, along with different protease inhibitors (Supplementary Fig. 5a). The stability of UmPR-1La proteins was enhanced in the presence of the cysteine protease inhibitor E64. Increasing the amount of E64 effectively blocked the cleavage of UmPR-1La (Fig. 5a), indicating that UmPR-1La is potentially cleaved by a plant cysteine protease.

As papain-like cysteine proteases serve as the central hub for apoplastic immunity[34], we further focused our experiments on these proteases. Peptide AtCAPE9 derived from Arabidopsis PR-1 activates plant immunity[8] and is reported to be released by Xcp1[10]. In maize, homolog *Xcp2* (NP_001149806.1) is wounding stages/wounding stages (Supplementary Fig. 5b). To determine if Xcp2 cleaves at the CNYx motif, we performed cleavage assays on recombinant proteins tomato PR-1b, maize PRB1-3, and PR-1La, as well as on *U. maydis*-secreted UmPR-1La proteins (Supplementary Fig. 5c–e). After treating plant PR-1s and PR-1La with Xcp2, truncated fragments were detected using Coomassie-blue-staining-PAGE and immunoblot for PRB1-3 and PR-1b (Supplementary Fig. 5c). In the case of *U. maydis*-secreted UmPR-1La, Xcp2 likely cleaved at the N-terminus of UmPR-1La to produce three different sizes of truncated fragments (Supplementary Fig. 5d). To directly detect the peptides, the Xcp2-cleavage samples were trypsin-digested, followed by targeted LC-MS/MS analysis. However, we were unable to detect the short tryptic peptide PPGNFR corresponding to ZmCAPE (PPGNFRGQRPY) of maize PRB1-3 or the tryptic peptide PPGNYIGK derived from UmCAPE-La (PPGNYIGKFKENVSPN) (Supplementary Fig. 6a).

We next investigated the cleavage by another cysteine protease, *Cathepsin B-like 3* (*CatB3*; ONM57676.1), which is expressed in the early stage of infection and strongly induced from 4dpi and onwards (Supplementary Fig. 5b). When treated with CatB3, the intensity of full-length UmPR-1La proteins was reduced but no truncated fragments were detected, indicating a different cleavage pattern compared to Xcp2 (Fig. 5b, c and Supplementary Fig. 5d). This suggests that cleavage occurred specifically at the C-terminus, potentially generating UmCAPE-La peptides. The activity of CatB3 was blocked by E64 (Fig. 5b), and alanine substitutions at either Cys121 or both Cys121 and His276 of CatB3's catalytic active site[35] prevented the cleavage of UmPR-1La (Fig. 5c). In the case of plant PR-1s, we surprisingly discovered that CatB3 cleaved recombinant PR-1La but not plant PR-1 proteins (Fig. 5d). In line with Fig. 5d, the ZmCAPE-derived short tryptic

peptide PPGNFR was not detected in the CatB3-cleavage PRB1-3 sample by targeted LC-MS/MS analysis (Supplementary Fig. 6b). However, the short tryptic peptide PPGNYIGK of UmCAPE-La was consistently identified in two independent experiments using CatB3-cleaved recombinant PR-1La and UmPR-1La samples (Fig. 5e; Supplementary Fig. 6b). These findings suggest that CatB3 specifically cleaves after the CNYD motif of UmPR-1La, resulting in the release of UmCAPE-La peptides.

### *U. maydis* deploys UmCAPE-La to suppress plant immunity

To determine the functional importance of UmCAPE-La in modulating plant immunity to promote fungal virulence, we analyzed the expression of maize immune-related genes during the early stages of *U. maydis* infection. qRT-PCR analysis revealed that the expressions of plant *PR-1*, *PR-2*, and *PR-5* genes were significantly induced upon infection by the ΔΔ*pr-1lab* strain compared to the wild type SG200, suggesting an additional role of UmPR-1L in subverting plant immunity (Fig. 6a).

We next investigated the role of UmCAPE-La peptides in regulating plant immunity by assessing disease development and defense gene expression in maize seedlings co-inoculated with ΔΔ*pr-1lab* and synthetic peptides. The application of UmCAPE-La peptides enhanced susceptibility to the ΔΔ*pr-1lab* infection, and conversely suppressed defense-related gene expression (Fig. 6b, c), suggesting the specific role of UmPR-1La in promoting fungal virulence by suppressing plant immunity via UmCAPE-La peptides. In contrast, the application of ZmCAPE peptides did not effectively reduce disease symptoms of ΔΔ*pr-1lab* compared to the mock control (Fig. 6b). This contrasts with previous studies showing a boost in plant SA-dependent immunity by CAPE peptides[8,9], which could potentially be attributed to the timing or method of ZmCAPE delivery.

To investigate this further, we generated the complementation strains replacing UmPR-1La's CAPE-La sequence with either the UmPR-1Lb's CAPE-Lb or ZmCAPE sequence (Fig. 6d). The Δ*pr-1la*_PR-1La(CAPE-Lb) strain, carrying UmCAPE-Lb peptides, partially restored the virulence of Δ*pr-1la*, resulting in an intermediate phenotype that showed no significant difference from SG200 or Δ*pr-1la*. In comparison to Δ*pr-1la*_PR-1La(FY*) cells which could not initiate hyphal-like structures and restore the virulence of Δ*pr-1la*, this finding suggests that shifting to hyphal growth is the primary function of UmPR-1La. Conversely, Δ*pr-1la*_PR-1La(ZmCAPE), with ZmCAPE peptides, significantly reduced the *U. maydis* virulence, leading to the development of tiny tumors with a size of less than 1 mm. This suggests a role of ZmCAPE in plant defense, and the spatial and temporal delivery of ZmCAPE peptides by *U. maydis* could effectively boost plant immunity to suppress fungal virulence. The substantial decrease in virulence caused by Δ*pr-1la*_PR-1La(ZmCAPE) could be partly rescued by applying UmCAPE-La, but not by a scrambled peptide of UmCAPE-La (Fig. 6e). These findings highlight the contrasting roles of *U. maydis* CAPE-La and plant CAPE in regulating plant immunity.

## Discussion

In this study, we have uncovered the multifaceted functions of UmPR-1La in *U. maydis* adaptation and virulence. The presence of a Ser/Thr-rich N-terminal region enables its localization on the cell surface, facilitating the C-terminal CAP domain mediated detection of environmental signals such as phenolics. This sensing mechanism helps fungal cells withstand environmental stress and initiate a shift from vegetative to filamentous growth, which is crucial for their infection strategy and survival within the plant host. Furthermore, the cell-surface projection of the CAP domain facilitates the release of CAPE-L peptides by CatB3 to suppress plant immunity and promote fungal growth and colonization (Fig. 7). Our work highlights that UmPR-1La is a versatile protein that plays a pivotal role in the complex interplay between fungi and their plant hosts. Its ability to present at the cell

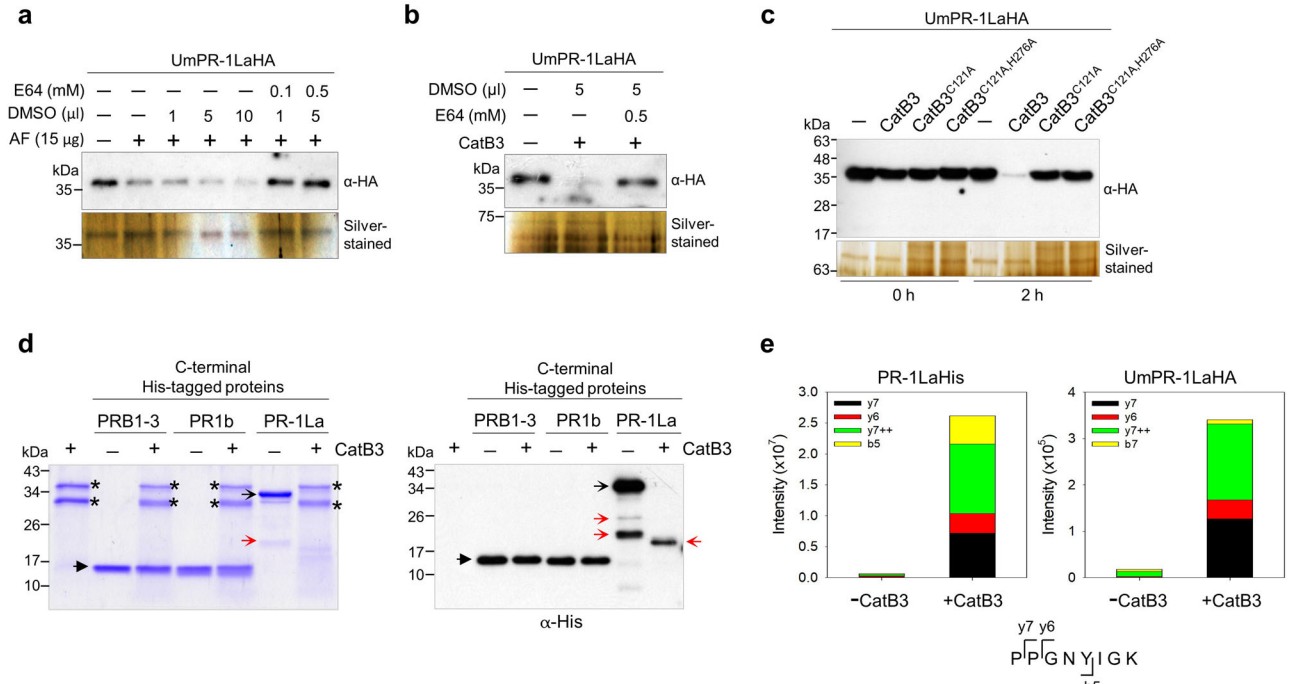

**Fig. 5 | UmCAPE-La peptide is released from UmPR-1La by CatB3. a, b** E64 blocks the cleavage of PR-1La. The SG200_PR-1La culture supernatant (labeled as UmPR-1LaHA) was incubated: (**a**) with the apoplastic fluid (AF) proteins from SA-inoculated maize, along with varying amounts of DMSO-dissolved E64 or an equivalent volume of DMSO, (**b**) with CatB3 in the presence of either 5 μl of E64 (final conc. of 0.5 mM) or DMSO. A representative blot from one of two independent experiments is shown. **c** The catalytic-site mutants of CatB3 fail to cleave UmPR-1La. The immunoblot showed the results before and after 2-hr incubations of SG200_PR-1La culture supernatant with CatB3 or its mutants. CatB3$^{C121A}$: the alanine substitution at Cys121. CatB3$^{C121A H276A}$: the alanine substitutions at Cys121 and His276. A representative blot from one of two independent experiments is shown. The loadings were verified using separated silver-stained PAGEs. **d** CatB3 does not cleave plant PR-1s. Maize PRB1-3 and tomato PR1b (C-terminal His-tagged proteins), and maize CatB3 (no tag) were purified from the apoplast of *N. benthamiana*. Full-length PR-1LaHis proteins was purified from *E. coli*. The CatB3-digested proteins

were separated on SDS-PAGE and stained with Coomassie-blue or analyzed by immunoblotting. Asterisks: the CatB3 protein bands. Black arrowheads: plant PR-1 proteins. Black open arrows: PR-1La proteins. Red open arrows: truncated PR-1La proteins. A representative blot from one of three independent experiments is shown. **e** Detection of UmCAPE-La peptides through targeted LC-MS/MS analysis. The SG200_PR-1La culture supernatant or recombinant PR-1La protein was treated with CatB3, as described in Fig. 5c, d, before subjected to LC-MS/MS analysis. The quantity of short tryptic peptides (PPGNYIGK) was measured in both the -CatB3 and +CatB3 samples using the peak area of fragment ions at specific retention times shown in Supplementary Fig. 6b. A cartoon illustrates the fragment ions of the peptide. Fragment ion b5 extends from the N-terminus, and y6 and y7 ions extend from the C-terminus, while y7++ is a doubly charged ion. The term "intensity" refers to the amplitude of the free induction decay signal. Consistent findings were observed in two independent CatB3-cleavage experiments using recombinant PR-1LaHis and UmPR-1LaHA. Source data are provided as a Source Data file.

surface, sense environmental stimuli, integrate signals for adaptation, and manipulate host immunity makes it a crucial factor in fungal adaptation and virulence.

Plants and their fungal pathogens possess the ability to recognize each other and activate complex signaling reactions in both organisms. These fungi rely on sensing and integrating environmental cues to initiate the yeast-to-hyphae switch and direct hyphal growth in plants[36–40]. This transition likely increases contact surface with plant cells, facilitating nutrient uptake. The deletion of *PR-1La* does not completely hinder *U. maydis* hyphal formation or fully attenuate disease progression, indicating that fungal cells rely on multiple distinct signaling cascades to initiate filamentous growth within the host, a critical step in fungal pathogenesis.

However, the yeast-to-hyphae transition could be disrupted by plant phenolics, such as ferulic acid[41]. These phenolics, which are the building blocks of lignin, are induced during infection[42]. Breaking down of lignin causes a release of phenolics and leads to their accumulation at infection sites to serve as defense agents to eliminate pathogenic intruders[43]. While the exact origins of *PR-1L* genes in fungal pathogens remain unclear, their capacity to act as substrates for plant proteases and conservation at CAP domain residues underscore a related role in plant-pathogen interactions during the process of co-evolution. The *PR-1-like* genes evolved to develop new functionalities to confront toxic phenolic compounds encountered during the

interactions and transmit signals to trigger hyphal branching inside the host plants. *U. maydis* hyphae usually orient their growth direction towards maize vascular bundles where they access carbon sources[44,45]. Given that lignin is enriched in the cell walls of maize vascular bundle sheaths, xylem, and sclerenchyma[46], *U. maydis* may sense phenolic compounds in addition to nutrient gradient[22] by a means to adjust hyphal growth and direct it toward vascular bundles.

Although ScPRY1 can bind to eugenols, its inability to induce hyphal formation hampers the rescue of *U. maydis* sporidial cells from toxicity. Unlike ScPRY1, which localizes to the cell periphery of *S. cerevisiae* to shield cells from eugenols, UmPR-1La accumulates at the cell division site of *U. maydis* sporidial cells. This localization pattern provides insufficient protection against eugenols. However, this situation could change upon transitioning to hyphal growth, enabling UmPR-1La binding facilitated by ligands on the hyphal wall and increasing the exposure of CAPE-La for cleavage by CatB3. Currently, the intracellular transduction of the signal perceived by UmPR-1La to induce the transition remains unclear. We speculate that phenolic binding induces a conformational change, which could be detected by an unidentified UmPR-1La partner, activating the downstream signaling pathway. Despite Sr10279 possessing conserved aromatic residues in the CBM and the ability to attach to *U. maydis* hyphal wall, it falls short of replacing UmPR-1La's protective function. We hypothesize that Sr10279 might not interact with the UmPR-1La partner. The

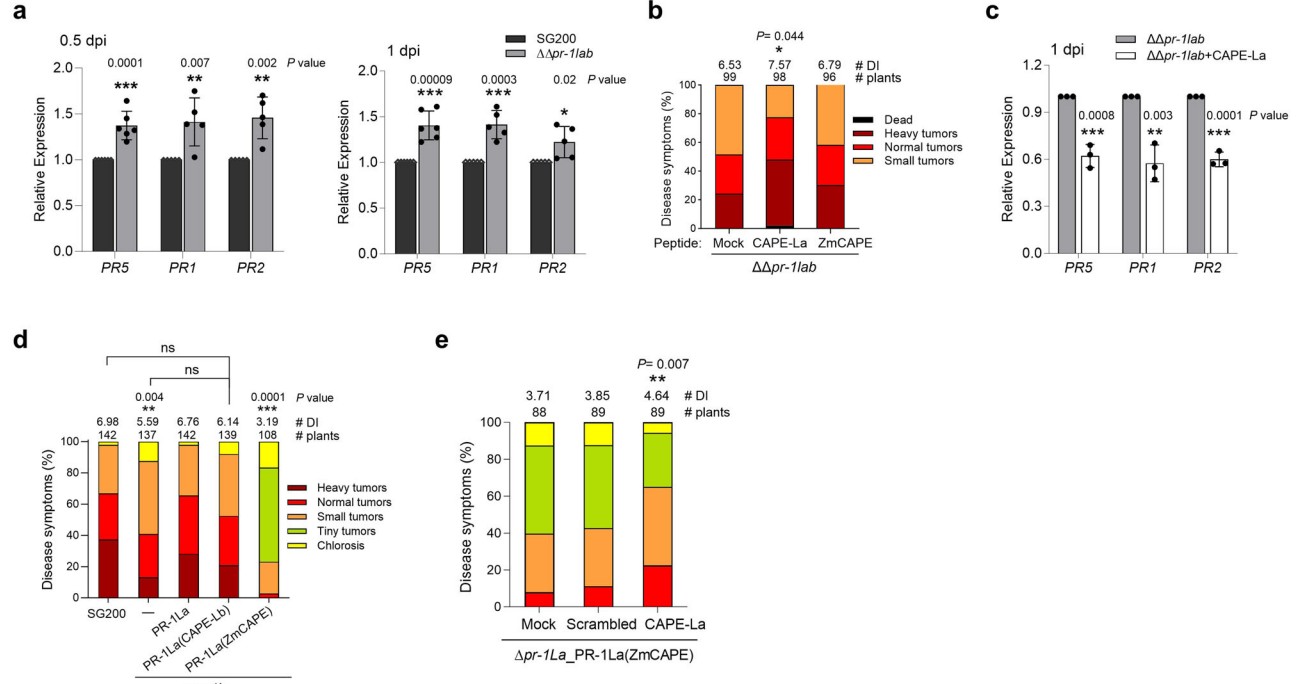

**Fig. 6 | UmCAPE-La enhances plant susceptibility by suppressing plant response. a**, **c** Relative expression of maize *PR* genes at early infection. Maize seedlings inoculated with (**a**) SG200 and ΔΔ*pr-1lab* without any peptides, or (**c**) ΔΔ*pr-1lab* with 0.2 μM synthetic peptide CAPE-La (PPGNYIGKFKENVSPN). RNAs prepared from infected leaves harvested at indicated time point were subjected to qRT-PCR analysis. Expression levels of *PR* genes were normalized to the *GAPDH* expression. The normalized *PR* gene expression in (**a**) SG200 or (**c**) ΔΔ*pr-1lab* was set to 1. Values represent mean ± sd of three independent infections. Significant differences between two samples were determined by a two-tailed unpaired Student's *t* test (**P* < 0.05; ***P* < 0.01; ****P* < 0.001). *PR1* (NM001147273); *PR2* (HM021761); *PR5* (U82201). **b** UmCAPE-La enhances the ΔΔ*pr-1lab* virulence. Maize seedlings were inoculated with ΔΔ*pr-1lab* along with 0.2 μM CAPE-La or ZmCAPE (from ZmPRB1-3). Disease symptoms were scored at 12 dpi. Total numbers of infected plants and the average DI determined from three independent infections are indicated above the respective columns. Statistical significance relative to the mock control were assessed using a two-tailed unpaired Student's *t* test. **d** *U.*

*maydis*-delivered ZmCAPEs suppress fungal virulence. Δ*pr-1la*_PR-1La(CAPE-Lb) and Δ*pr-1la*_PR-1La(ZmCAPE): Δ*pr-1la* expressed the PR-1La mutant proteins carrying UmCAPE-Lb (PAGNVEGLFDAQVPAKVQPTPRLRSSCSANERHGS) or ZmCAPE peptides under the control of *PR-1La* promoter. The indicated strains were inoculated into maize seedlings. A description of each disease category is provided in the Method section. The average DI values determined from at least three independent infections are shown. Significant differences between SG200 and the indicated strains were determined by a two-tailed unpaired Student's *t* test. Δ*pr-1la*_PR-1La(CAPE-Lb) showed no statistically significant (ns) differences compared to the indicated strains. **e** UmCAPE-La mitigates the effect of ZmCAPE on fungal virulence. Maize seedlings were inoculated with Δ*pr-1la*_PR-1La(ZmCAPE) along with 0.5 μM of CAPE-La, the scrambled peptide of CAPE-La (YFSKNIKNPPVEGNGP), or water (mock). The average DI values determined from three independent infections are shown. ***P* < 0.01 denotes significant difference between the CAPE-La and mock inoculation, determined by a two-tailed unpaired Student's *t* test. Source data are provided as a Source Data file.

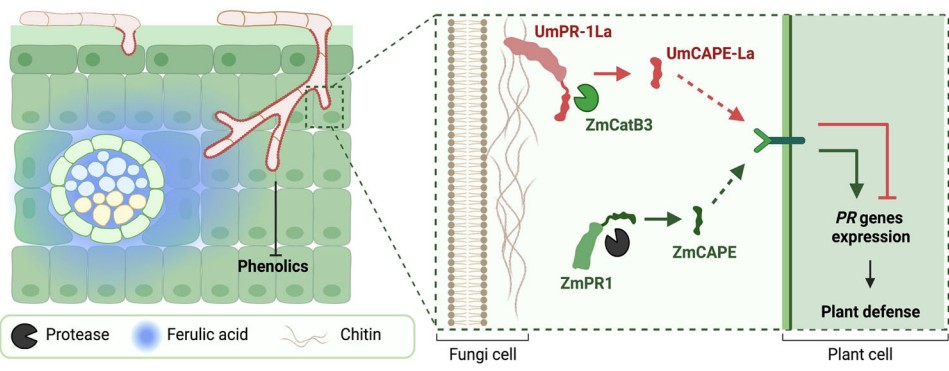

**Fig. 7 | Hypothetical model for the PR-1La function.** Upon host entry, *U. maydis* employs UmPR-1La to sense phenolics, initiating signals for shielded hyphal branching as a defense against toxic phenolics. This sensing mechanism promotes fungal hyphal growth towards phenolic-enriched vascular bundles. Additionally, the exposure of UmCAPE-La on the cell surface facilitates its cleavage by CatB3 and subsequent release. UmCAPE-La presumably competes with ZmCAPE for binding to an unidentified receptor, suppressing plant immunity and enhancing *U. maydis* virulence. Created with BioRender.com.

potential interacting motif possibly resides in the highly diverse and disordered N-terminal region, with functions that extend beyond its role in facilitating PR-1La protein cell wall localization. Further investigation is necessary to comprehend how N-terminal regions

collaborate with CAP domains to participate in the UmPR-1La-dependent signaling pathway.

The Δ*gpa3* and Δ*bpp1* mutants exhibit a similar pseudohyphal structure phenotype with UmPR-1La-expressing cells, possibly

suggesting the involvement of Gα and Gβ proteins from the cyclic AMP-dependent protein kinase A (cAMP/PAK) pathway[47,48]. However, given that PR-1La lacks a transmembrane domain, it is not clear how the cell surface signals would be perceived and transduced to activate downstream signaling. Several studies have reported the discovery of G proteins in extracellular vesicles (EVs), and that GPCRs regulate EV release to mediate intercellular communication between neighboring and distant cells[49–51]. The finding that EVs trigger the yeast to hyphal transition in *Candida albicans*[51] also supports the possibility that EVs containing signaling proteins such as GPCRs could be involved in transmitting the UmPR-1La signal to switch on filamentous growth. Further study is needed to explore cellular signaling switching to filaments.

Maize Xcp2 and CatB3 are papain-like cysteine proteases that play an important role in apoplastic immunity[34]. This explains why this group of proteases is targeted by pathogens. For example, the *U. maydis* effector Pit2 inhibits the activity of maize cysteine proteases CP1, CP2, and Xcp2 to prevent the release of Zip peptides and subsequent inactivation of plant SA-dependent signaling[52,53]. However, Pit2 does not inhibit CatB3. In addition, *U. maydis* also induces the maize cystatin ZmCC9 to inhibit the activities of cysteine proteases, but it has a comparatively weak interaction with CatB3[54]. This evidence, and our observation that CatB3 releases UmCAPE-La, highlights a sophisticated strategy where *U. maydis* suppresses plant immunity by hijacking maize CatB3 to release a potential functional mimic of ZmCAPE peptide from PR-1La. UmCAPE-La may compete for binding to a yet unknown receptor, leading to inactivation.

It is unclear why CatB3 avoids cleavage of plant PR-1. One speculation is that CatB3 could not access the motif unless a conformational change is induced in PR-1s, which is possible by a specific ligand binding to the PR-1 CAP domains. Alternatively, an unidentified protease is responsible for plant CAPE release from PR-1s. Although AtXcp1 is reported to cleave the CNYD motif and release AtCAPE9 peptides[10], maize Xcp2, CatB3, and wheat serine protease fail to release monocot CAPEs[9]. Nonetheless, our work opens a new avenue to understand how CatB3 excludes plant PR-1 as its target and whether another protease is responsible for releasing CAPEs in monocots.

Our work provides a mechanistic bridge between understanding how fungal pathogens employ a conserved CAP-domain PR-1L protein to perform a function contradictory to the defense function of PR-1. In the future, identifying the receptor of plant CAPE and fungal CAPE-L peptides will provide more insights into the activation of plant immunity via CAPE-dependent signaling cascades. This will further solve the unresolved questions concerning the role of PR-1 proteins in plant defense.

# Methods

## *U. maydis* strain construction, growth conditions, and virulence assays

*Zea mays* The *U. maydis* infection was performed using the Honey 236 Taiwan cultivar. The haploid solopathogenic *Ustilago maydis* SG200 strain was used as a reference strain in this study[25,55]. All *U. maydis* strains generated and used in this study are listed in Supplementary Table 1. *U. maydis* strains were cultured on solid potato dextrose agar (PDA) plates (2.4% potato dextrose broth and 2% agar). AB33 derivatives were grown in YEPSL liquid medium (0.4% yeast extract, 0.4% peptone, and 2% sucrose) until reaching an $OD_{600}$ of 0.2 at 28 °C, Subsequently, they were shifted to grow in nitrate minimal medium (NM)[56] supplemented with 2% glucose for 8 h to induce *b*-dependent filaments. For infection, strains were grown in YEPSL liquid medium until reaching an $OD_{600}$ of 0.8 at 28 °C. Cells were adjusted to a final $OD_{600}$ of 1.0 in $H_2O$ and injected into stems of 7-day-old maize seedlings. Disease symptoms were assessed and scored at 12 days post-infection (dpi) using the disease rating criteria described in the

previously report[25]. The disease index (DI) was calculated as the sum of values from all disease categories, which was obtained by multiplying the number of plants in each category by their corresponding assigned value (Death plant = 11; Heavy tumors on base of stem or stunted growth = 9; Normal tumors (>3 mm) on leave and/or stem = 7; Small tumors (between 1-3 mm) = 5; Tiny tumors (<1 mm) = 3; Chlorosis = 1), and then dividing by the total number of infected plants[57]. Significant differences in disease symptoms between each strain and SG200 or the indicated strain are determined using a two-tailed unpaired Student's *t* test.

## Plasmid and strain construction

Plasmid construction using either Gibson Assembly or standard cloning methods are described in Supplementary Table 2. Primers used in each generated plasmid are listed in Supplementary Table 3. A PCR-based approach was used to generate mutants[58]. For gene integration into the *ip* locus, plasmids containing a carboxin resistant *ip* allele (*ip*R)[59] were linearized with restriction enzymes and subsequently inserted via homologous recombination. The transformation of *U. maydis* and genomic DNA isolation procedures were performed as described in a previous study[60]. Positive *U. maydis* transformants were verified by Southern blot analysis.

## Gene expression analysis

Total RNAs from infected maize leaves were extracted using QIAGEN RNeasy Plant Mini Kit (Qiagen #74904), followed by DNase-treatment and reverse-transcription before qRT-PCR analysis[61]. The genomic DNA was removed from the total RNA using a TURBO DNA-free™ Kit (Invitrogen# AM1907). The cDNA was reverse-transcribed using a SuperScript® III First-Strand Synthesis SuperMix (Invitrogen# 18080400). The expression of maize *GAPDH* was used for normalization. The relative expression values of *genes* were calculated using the $2^{-\Delta\Delta Ct}$ method[62]. Quantitative real-time PCR analysis was performed using primer pairs for indicated genes listed in Supplementary Table 3.

## Immunolocalization of PR-1La

Filamentous cells were induced by spraying sporidial cells with hydroxyl fatty acids on parafilm, as described in the previous report[44]. Parafilms with attached filamentous cells were then incubated with or without 5 μg of purified proteins in 2 ml PBS buffer (pH 7.4) for 4 h at room temperature. Subsequently, immunostaining was performed using anti-His/anti-HA antibody (Yao-Hong Biotech., Taiwan; #YH80003 and #YH80007; 1: 2000 dilution) and Alexa Fluor 488/594 (AF488/594)-conjugated secondary antibody (Invitrogen #A28175; Abcam # AB150116; 1: 2000 dilution)[44]. For the colocalization study in AB33 filaments, the AF594-labeled filaments were stained with 1 μg/ml of wheat germ agglutinin-conjugated Alexa Fluor 488 (WGA-AF488; Invitrogen# W11261) for 10 min in a PBS buffer. The cells were then subjected to a 1-min vacuum in a PBS buffer containing 1 M NaCl before being visualized using microscopy. The fluorescence was observed using Axio Observer fluorescence microscope equipped with Axiocam 702 Monochrome camera (ZEISS, Germany). Images were processed using ZEN 3.2 imaging software (ZEISS).

## Protein purification

Plant PR-1His, Xcp2-His, and CatB3 proteins, which contain a signal peptide, were expressed in *Nicotiana benthamiana* using Agroinfiltration method[63]. *Agrobacterium tumefaciens* GV3101 strains carrying expression plasmids were infiltrated with needleless syringe to the leaves of 3-weeks-old *N. benthamiana* plants. To purify CatB3 proteins from the apoplastic fluid, infiltrated leaves harvested at 3 dpi were immersed in an infiltration buffer (25 mM Tris-HCl, 250 mM NaCl, pH 7.5), vacuum-infiltrated for 5 min, and then centrifuged at 1000 g for 5 minutes to collect apoplastic fluid. The apoplastic fluid was concentrated and exchanged with buffer A (25 mM Tris-HCl, 150 mM NaCl,

pH 7.5) using 10 kDa cutoff filters before being subjected to size-exclusion chromatography. Fractions containing CatB3 proteins were concentrated and kept in buffer A containing 10 % glycerol. To purify PR-1His and Xcp2His proteins, the apoplastic fluids were exchanged with buffer B (25 mM Tris-HCl, 250 mM NaCl, 15 mM Imidazole, pH 7.5) before being incubated with Ni-NTA agarose beads. The beads were washed with the infiltration buffer, and His-tagged proteins were eluted, buffer-exchanged, and kept in buffer A containing 10% glycerol.

Recombinant His-tagged full-length or truncated PR-1La protein was expressed in *E. coli* Shuffle T7 cells (NEB #C3026J). To purify recombinant proteins, *E. coli* cells were disrupted in a lysis buffer (25 mM Tris-HCl, 250 mM NaCl, 20 mM MgCl₂, 20 mM KCl, 20 mM Imidazole, pH 7.5) supplemented with 0.5% Triton X-100, 0.5 mg/ml lysozyme, 0.1 mg/ml DNase, and 1 mM PMSF before being subjected to Ni-NTA affinity purification as described above, except the beads were washed with the lysis buffer containing 30 mM Imidazole. The eluted proteins were further subjected to size-exclusion chromatography. Purified proteins were kept in buffer A containing 10% glycerol.

### Fluorescence spectroscopy

In a final volume of 2 ml of buffer containing 50 mM sodium acetate at pH 5.5, the recombinant PR-1LaHis protein (0.25 μM) was mixed with phenolic compounds and incubated for 15 min at room temperature on a rotating wheel. This was followed by measuring the fluorescence intensity of the mixture using Jasco fluorescence spectrophotometer FP-8300 with Spectra Manager software (version 2). The excitation wavelength was set at 290 nm to selectively excite tryptophan residues of PR-1La, and the emission spectra were recorded in the wavelength range of 310–450 nm with a 1 nm interval, using a scan speed of 100 nm/min.

### Protease inhibition assay

A total of 30 μg of apoplastic fluid proteins from SA-treated maize leaves were mixed with 3 μl of concentrated culture supernatant from *U. maydis* cells expressing PR-1LaHA proteins in a 50 μl reaction buffer (50 mM sodium acetate and 10 mM DTT, pH 5.5). The reaction mixture was incubated in the presence of 0.1 mM protease inhibitor (E64, Sigma #E3889; Pepstatin A, Sigma #P5318; 3,4-Dichloroisocoumarin, Sigma #D7910; cOmplete protease inhibitor cocktail, Roche), 1 mM EDTA (Sigma# E9884), or 1 μl of DMSO control at 28 °C for 2 h.

### Protease cleavage assay

A total of 2.5 μg of CatB3 or 1 μg of Xcp2 protein was pre-activated in a buffer containing 50 mM sodium acetate (pH 5.5) and 10 mM DTT for 30 minutes at 28 °C. Subsequently, 2.5 μM of purified PR-1 or PR-1La protein, or 2 μl of concentrated culture supernatant from SG200_PR-1La, was added. The reaction mixture was further incubated at 28 °C for an additional 30 minutes (for Xcp2-cleavage) or 2 h (for CatB3-cleavage), respectively. The protein cleavage pattern was analyzed by western blotting using mouse anti-His (1:8000; Yao-Hong Biotech., Taiwan; #YH80003) or mouse anti-HA antibody (1:8000; Yao-Hong Biotech., Taiwan; #YH80007) and goat horseradish peroxidase-conjugated anti-mouse as secondary antibody (1:25,000, Yao-Hong Biotech., Taiwan; #AS111772).

### Phenolic treatment

Cells with an initial OD of 0.1 were grown in YEPSL liquid medium containing 1 mM eugenol (Sigma # E51791), 7 mM ferulic acid (Sigma #128708), 5 mM coniferyl alcohol (Sigma #223735), or 5 mM coumaric acid (Sigma #C9008) at a final pH of 6.0, and then the cells were incubated at 28 °C for up to 48 h.

### Sample preparation for LC-MS/MS

Protease-cleavage samples were extracted using the suspension trapping (S-Trap) protocol[64]. Briefly, the samples were lysed in a buffer containing 50 mM Tris-HCl (pH 8.0) and 5% SDS, and the protein concentrations was determined using the BCA protein assay (Thermo Fisher Scientific #23225). Before trypsin digestion, disulfide bonds within proteins were reduced with 10 mM Tris(2-carboxyethyl)phosphine hydrochloride and alkylated with 40 mM 2-chloroacetamide (CAA) at 45 °C for 10 min. Each protein sample was digested into peptides by Lys-C and trypsin in an enzyme-to-protein ratio of 1:50 (w/w) within an S-trap micro microcolumn. Following digestion, the peptides were eluted using 40 μl of a solution composed of 50 mM triethyl ammonium bicarbonate, 0.2% formic acid (FA), and 50% (v/v) acetonitrile (ACN). Subsequently, a desalting step was conducted using a Ziptip Pipette Tip and the desalted peptides were dried using a vacuum centrifugation concentrator. The dried peptides were reconstituted in 0.1% (v/v) FA, centrifuged to collect the supernatant, and finally subjected to LC-MS/MS analysis.

### LC-MS/MS analysis

For enhanced confidence in peptide identification during LC-MS/MS analysis, synthetic peptides with stable heavy isotope-labeled lysine or arginine were spiked into the dried peptides, and then dissolved in 10 μl of 0.1% FA. The peptide analysis was conducted using Thermo Fisher Scientific Q-Exactive mass spectrometer coupled with an Ultimate 3000 (Thermo Fisher Scientific), controlled by Xcalibur software (version 4.3.73.11). A nanoEase M/Z Peptide CSH C18 column (75 μm x 25 cm, 1.7 μm, 130 Å) with a column heater set at 45 °C was employed for peptide separation. Peptides separation was achieved through a linear gradient of 5–25% (Buffer A consisted of 0.1% FA, and buffer B contained 0.1% FA in 100% CAN) over a 60-minute duration, at flow rate of 300 nl/min. The acquisition cycle of the MS data was executed in parallel reaction monitoring (PRM) mode, with a full survey MS scan followed by the mass-to-charge (*m/z*) range 300–1600. The MS scan was performed with a resolving power of 70,000. The PRM MS/MS acquisitions were performed using a 2.0 Da isolation window, a maximum 250 ms ion injection time, 27% NCE (normalized collision energy), and a resolving power of 35,000.

### PRM data analysis

The PRM raw files were analyzed using Skyline (version 22.2)[65]. The fragment spectra predictions for PPGNYIGK and PPGNFR was conducted using Prosit[66]. Protein digestion was set to Trypsin (semi) [KR|P], with a minimum length of peptide set to 6 amino acids. Variable modifications applied included Label:13 C(6)15 N(2)(C-term K) and Label:13C(6)15N(4)(C-term R). Mass accuracy filtering was set to 10 ppm. Ion match tolerance was set at 0.5 m/z, and a minimum of 3 product ions was required for analysis. Both fragment b and y ions were employed for quantification. The peak area of selected fragment ions was extracted for quantification. All data were checked manually for peak selection and retention time.

### Data and bioinformatic analyses

Data relevant for *CatB3 and Xcp2* gene expression analysis were extracted from the RNAseq dataset available at NCBI Gene Expression Omnibus under accession number GSE103876[23]. To perform phylogenetic analysis, the amino acid sequences of PR-1 and PR-1-like proteins are retrieved from the NCBI database, aligned using Clustal Omega[67], and trimmed by using trimAl (version 1.4.1)[68]. Evolutionary analysis was performed in MEGA 7.0[69], and the phylogenetic tree was constructed using Maximum Likelihood method with the WAG + G + I model and 1000 bootstrap replicates. MEME (version 5.5.2)[70] was used to obtain the consensus sequence of CAPE and CAPE-like peptides. The scrambled peptide containing an identical amino acid composition to UmCAPE-La, is randomly generated using Scrambled tool in Mimotopes website.

**Reporting summary**

Further information on research design is available in the Nature Portfolio Reporting Summary linked to this article.

## Data availability

MS raw data files for targeted MS analysis have been deposited to the ProteomeXchange Consortium via the PRIDE partner repository, under the dataset identifier PXD044915. Unprocessed data, gels, and blots are provided in the Supplementary Information and the Source Data file. Source data are provided with this paper.

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

## Acknowledgements

We appreciate Regine Kahmann for providing the Δpr-1la and Δpr-1lab mutants (Import permit# 109-F-500 and 107-F-002). We acknowledge Drs. Paul Verslues, Tien-Shin Yu, and Erh-Min Lai for providing their valuable suggestions and comments. We thank the staff from the Proteomics Core Laboratory and DNA Sequencing Core Laboratory at the Institute of Plant and Microbial Biology for technical assistance. This work was done in collaboration with Drs. Yin-Ru Chiang and Chuan-Chih Hsu and is supported by grant from Academia Sinica Thematic Research Program (#AS-TP-111-L01).

## Author contributions

L.S.M. conceived and designed this study. L.S.M. and Y.R.C. wrote the manuscript with input from all authors. Y.H.L., M.Y.X., and F.A.D. performed immunolocalization. Y.H.L., M.Y.X., and H.C.L. performed phenolic treatment assays. Y.H.L. and M.Y.X. did the protein purification. WLT did fluorescence spectrometry analysis. Y.H.L. did RT-PCR analysis and protease cleavage assays. C.C.H. performed LC-MS/MS analysis. Virulence assays were done by M.Y.X., F.A.D., and C.V.H. All authors discussed the results and commented on the manuscript.

## Competing interests

The authors declare no competing interests.
