## [Peer Review File · Nature Communications]

REVIEWER COMMENTS

Reviewer #1 (Remarks to the Author):

PR-1-like proteins belong to a large protein family, and they have been implicated in many different biological processes including immune defense and cancer progression. These proteins share a conserved CAP domain and are mostly secreted glycoproteins. Their mode of action, however, remains poorly characterized. Thus, progress in understanding the function of these proteins at a molecular level would be a welcome advancement.

Lin et al. now report that the corn smut *Ustilago maydis* pathogenesis-related 1-like protein (UmPR-1La) acts in detoxification of phenolic compounds, and that its Ser/Thr-rich N-terminal domain mediates cell surface association. In addition, the authors report that UmPR-1La has gained a specialized activity in eliciting hyphal formation and they suggest that the protein senses phenolic compounds to direct hyphal growth in plants. The third reported finding is that UmPR-1La is cleaved by a plant cathepsin B-like protease to release a functional signaling peptide (CAPE-like) that suppresses plant immunity and promotes fungal virulence.

The main limitation of this study is that it presents an assembly of three more or less independent observations but fails to convey a compelling and significant overall progress in our understanding of the mode of action of PR-1-like proteins. Moreover, each of the three findings is not sufficiently supported by data.

Specific comments:

Fig1a, the consensus sequence for the CAPE1 cleavage is CNYx. It is not clear why the authors change this consensus to CxYx. The same applies for the 11-amino acid CAPE-like peptide sequence. The plant PR-1 derived CAPE1 peptides all end in R/KPY but this is not conserved in UmPR-1L. It is not clear what criteria these authors apply to define the CAPE-like peptide derived from UmPR-1La as being similar/like to that of the plant CAPE1 peptides.

Fig1b, the normalized read counts for pr-1a appear to be very low, especially when compared to those for pr-1b, which are 100-fold higher (FigS1b), or CatB3, which are about 500-fold higher (Fig5a). The gene thus appears to be expressed only at very low levels.

Fig1d, are the disease symptoms between the single mutant (Δa) and the double mutant ($\Delta\Delta ab$) statistically significant? The authors describe these differences as “more pronounced” (L137).

Fig2a,b, the authors say that UmPR1-La and ScPry1-mCherry are cell surface associated proteins, and that UmPR1-La localizes to bud necks and growing tips in sporidial cells. However, they do not show any colocalization experiments with established cell wall markers. At the same time, they show that these proteins are secreted, but they do not include a cell wall-associated protein in the Western blots as control. Are these proteins now secreted or are they cell wall associated?

Fig2e, shows that recombinant full length and the Ser/Thr-rich domain of UmPR1-La bind chitin and chitosan. The authors do not show whether these proteins are soluble under these conditions in the absence of chitin or chitosan. If the Ser/Thr-rich domain would promote protein self-association, it would appear as binding to chitin/chitosan. These proteins were expressed and purified from bacteria, nevertheless, the authors suggest that they associate with the fungal cell wall by binding to chitin/chitosan via glycosidic linkages (L169-172). Does the CAP domain bind lignin and is this abrogated in a CBM (FY*) mutant version? This seems to be important given that the authors subsequently show that the CAP domain binds phenolic compounds that are the precursors and building blocks of lignin.

Fig3, the apparent role of UmPR1-La in conferring eugenol resistance and the putative function of UmPR1-La in sensing phenolic compounds to signal pseudohyphal growth is based on cells that overexpress this protein, i.e., a SG200 wild-type strain expressing an additional copy of UmPR1-La from a constitutive promoter. Thus, this is a gain-of-function phenotype that may have little in common with the native function of this gene/protein. These crucial experiments need to be supported by showing that mutant cells lacking one (Δa) or both ($\Delta\Delta ab$) UmPR1-L are hypersensitive to eugenol, and that this sensitivity and hyphal growth can be complemented in the deletion strain by expression of one or both UmPR1-L from their respective native promoters. The putative function of the CBM should then be tested in this sensitized background and not in an overexpressing strain. Are the WT and FY* mutant proteins used in Fig3c and Fig4b expressed at comparable levels, please provide Western data.

Fig4, as in Fig3, the authors should make sure that they are not scoring an overexpression phenotype. Is an SG200 strain not expressing WT or FY* mutant version of UmPR1-La sensitive to ferulic acid, what is its morphology? The binding assay shown in Fig4c does not indicate whether binding is specific, concentration-dependent, and saturable. Could it be that the protein gets denatured in the presence of ferulic acid and that this is scored as “binding”? This also applies to Fig4b.

Fig5, the authors fail to detect the UmCAPE-La peptide in the apoplastic fluid of *U. maydis* infected leaves (L239). Hence there is no evidence that this peptide is actually produced in planta. Did they try infection with an overexpression strain? Fig5b, CatB3, which shows cleavage activity towards recombinant UmPR1-La, was purified from the apoplastic fluid of tobacco leaves and this protein

preparation is not pure (FigS5b). Purification and cleavage assay with a catalytically dead mutant version of CatB3 would be needed to claim that the observed cleavage is indeed due to CatB3 and not a contaminating protease. This seems to be particularly important given that the cleavage of the native secreted version of UmPR1-L by CatB3 appears to be very slow and inefficient.

Fig6, if CAPE-L indeed has immune suppressive activity one would expect that *U. maydis* strains overexpressing UmPR1-La show enhanced virulence. What is the induction of PR gene expression when non-infected plants are treated with ZmCAPE, CAPE-L, or both peptides together, compared to plants treated with scrambled peptide sequences?

Reviewer #2 (Remarks to the Author):

The authors provide a functional characterization of 1 (2) effector(s) of the fungal maize pathogen *Ustilago maydis*, which have homology with plant PR1 proteins, including the C-terminal CAPE peptide which is known to play an important role in PR-1 mediated induction of plant defenses. The findings shown in the paper are new and exciting. Also, the experiments are all over sound. However, the presentation and documentation of data is partially incomplete. Also, this reviewer is not completely convinced how the two proposed functions of eugenol detoxification (which is actually rather resistance/tolerance than detox) and CAPE-release to block plant immunity are linked. Also, it appears unclear why/how the Um-PR1 released CAPE interferes with the generation of Zm-derived CAPE peptide.

General questions and comments:

- Are orthologs of related smuts able to complement the PR-1La disease phenotype, or is this specific to the *Ustilago maydis* protein?
- Does the PR-1La deletion mutant have any (growth/developmental) defects besides in-plant development? Given its function at cell-surface localization this seems to be relevant for the interpretation of the results and the actual biological function of the PR-1La.
- It appears (particular at level of this journal) somehow inappropriate to show gene expression data in the main figures (Fig 1; Fig 5) that is actually just extracted from a publication by others. Authors may provide own qPCR data to confirm the proposed expression patterns in their own experimental conditions (and refer to the published data in the supplements)

- The figure labels are missing some relevant information to make them accessible. In Fig 2 it took me a while to understand that I am looking on an “otef” OE which explains the signals in sporidia. In Figure 3C I needed to search for “FY*” and found it in the legend for 3D. However, as the figure 3C is shown, one would understand that there is a “WT” and a “FY*” both being somehow with “SG200_PR-1LaHA”. What I guess it is, is strain SG200 (which actually should not be called a “wild type”, because it is an artificial lab strain) vs the pr-1la deletion strain. Right?

Allover, it took me lots of time to look up this kind information up, as descriptions in the figures are sparse/sometimes misleading and also the legends lack information (e.g. legend Fig 3 “same strains used in 2a” – this is not sufficient. Please name the strains which are shown!) Thus, the authors should carefully revise their figures (including supplements, where also info is missing), provide more precise labels and information on strains wherever it is needed, otherwise it is very difficult to follow and interpret results.

- Does the CAPE-peptide have a virulence function for *U. maydis*? Figure 6 implies that it has a role in virulence. So, is a mutant with a C-terminally truncated PR-1La reduced in virulence? If so, can other (e.g. Zm) CAPE sequences not complement this?

- If *Ustilago* CAPE competes for Zm-CAPE with the receptor – how is it then not activating it (if it has a stronger binding affinity?)

Specific questions:

- In L136 it is stated that “double deletion phenotype is more pronounced” – Is this really significant compared to the single deletion?

- Figure 3 and corresponding text: what is the evidence for a “detoxification” – what is shown is resistance to a toxic compound, but not its detox.

- Figure 4C – what is actually detected here by the spectrophotometer? At which wavelength? Why are no biological reps shown but only technical reps?

- Figure 5 – also here, biological reps are not shown (MS data). Also, it is not possible for me to understand the figure completely. What are the sequences below 5b? What is shown in 5C (what is “y7” “y6” etc?) What is scale of “intensity”? How does this show a specific cleavage of PR-1La by CatB3?

- Fig S5A indicates degradation of PR-1La by apoplastic fluid of maize. However, none of the inhibitors used could fully block this degradation completely. This actually suggests that multiple proteases are involved in degradation of the protein beyond Cat3B. How do the authors interpret this?

Reviewer #3 (Remarks to the Author):

The submission, “Ustilago maydis PR-1-like protein has evolved two distinct domains for dual virulence activities” by Lin et al describes a fungal protein, PR-1-like, that has homology to the well described yeast, PRY1 and plant PR1. The authors show that the U. maydis PR-1-like proteins contains two domains, Ser/Thr-rich region and the CAP domain with a predicted C-terminal peptide similar to PRY1, while plant PR-1’s only contain the CAP domain with a C-terminal peptide. The authors show that the Ser/Thr-rich region is responsible for the binding of PR-1-like to Chitin and therefore its localisation to the fungal cell wall. While the CAP domain is responsible for sterol/phenolic binding which induces a switch in U. maydis from sporidia to pseudohyphal structures which allows survival of the fungi. Additionally, the authors show that the C-terminal peptide of PR-1-like is cleaved by CatB3 a member of the cysteine protease family and that PR-1-like can modulate known plant immune genes which is likely to occur via the C-terminal peptide. Overall, this paper is quite well written and logically structured making it relatively easy to follow the narrative, however some work on Figure layout and labelling will help greatly. I have below some major and minor points to address which should improve the article prior to publication.

Major point to address

- Figure 2 – These microscope images (a and d) are not good enough quality to be the only evidence of cell surface localisation. I would like to see better quality and higher resolution images as well as co-localisation with a known cell surface protein so show co-localisation. Arrows would also help to direct reader.
- Please review all figures as many are lacking good labelling e.g. Fig2, no labels on western blot for expected size of protein especially important for PR-1Lb as there are multiple bands and the band sizes don’t seem to match my calculations based on the amino acids described in fig 1. Also Fig2a - PRY1HA I believe should be labelled Umsp:PRY1-HA this was very unclear as the labelling did not match the description in the text of the results. These are just two examples.
- Add in the O-glycosylation data into a sup figure
- Figure 3b – Given that the authors base this experiment off the PRY1 literature of PR1 binding Eugenol this seems an unexpected result that the cells do not survive, please expand on/discuss this. Could it be linked to the data in Fig2a where PRY1 does not localise to cell surface in U. maydis? Make clear your reasons for reader.
- Add loading controls to immunoblot panels.

- I find it surprising not to have seen an enhancement of defence by the application of the maize CAPE peptide. I would suggest you repeat this experiment, but this time pre-treat the maize with ZmCAPE 1 day prior to infection. This experimental set up is how previous publications have conducted the experiment e.g. Chen et al, 2014 Plant cell and Sung et al, 2021 New Phyt (refs 8 & 9 on your list). Have you also tried it with WT SG200?

Minor point to address

- Line 79-81 “However, it is unclear whether a protease involves in the cleavage of the CNYx motif and a receptor to perceive CAPE peptide to activate” suggest editing to read “However, it is unclear whether a protease is involved in the cleavage of the CNYx motif and if there is a receptor to perceive CAPE peptides that activate”
- Line 95 – “sexual life cycle, it requires U. maydis fine sensing and integrating environmental” suggest editing it to “sexual life cycle, U. maydis requires fine sensing of environmental”
- Line 97 – “U. maydis secrete a” edit to “U. maydis secretes a”
- Line 108 – “elicit hyphal-like structures. A CAPE-like peptide” suggest edit to “elicit hyphal-like structures, while a CAPE-like peptide”
- Line 172-173 – I think this statement has not been 100% proven therefore I would say “the results indicates that this PR-1La may incorporate into the cell-wall chitin/chitosan-glucan matrix possibly via glycosidic linkages”
- Remind reader that PR-1-likes are not expressed in culture therefore the native PR-1-like’s are not affecting the phenolic binding assays.
- Figure S3a – there is wrong labelling between the graph and the images or there is a wrong image added – graph shows PR-1La showing cell survival and growth however image is showing PR-1Lb data. Need consistency between graph and images, suggest adding PR-1La and PR-1Lb to both.
- Line 193 – Wrong Figure, should read Fig 2a.

- Fig3d – suggesting adding in WT complementation line. Also inconsistent labelling between Fig 1d and 3d
- Line 218 – CA-treated PR-1La (FY*) cells show a swollen morphology, based on the image shown for PR-1La WT treated with CA these pseudohyphae also look swollen
- Inconsistent labelling of sterol graphs between Fig3 and 4. FigS3 and FigS4
- Lines 241-244 – re-word sentence, it is difficult to follow. You need to get the phrase culture filtrate or culture supernatant in there as well as saying U. maydis SG200 +PR-1La-HA line.
- Figure S5a legend is missing information – treatments and concentrations used. Labelling of figure and description in materials and methods don't match i.e. is control on figure the PIC as in materials you mention DMSO is the control?
- Line 249 – “Cathepsin B-like 3 (CatB3) has a similar expression profile as umpr-1a” I disagree PR-1La expression is all over the place. I would just say CatB3 is induced during infection or CatB3 is strongly induced from 4dpi compared to mock treatment.
- Fig5b add arrow to indicate PR-1La band also labels for Coomassie and immunoblot.
- Fig5b make the second blot on U. maydis secreted PR-1LaHA blot Fig 5c as currently the legend is hard to follow what is being described. Quantify band intensity against total protein stain as there is no size shift (due to small cleavage size) as observed in the previous gel and no loading controls shown.
- Line 265 – “In maize, homolog Xcp2 is highly expressed in the 266 early infection process and then slowly declines during biotrophy (Fig. S5d).” – I disagree with this statement, there is no difference in expression of Xcp2 between mock and infected tissue. Remove or reword.
- Line 268 – suggested edit “truncated fragments were detected by Coomassie-blue-staining-PAGE and immunoblot for PRB1-3 and PR-1b”

- Line 313-317 – “*U. maydis* probably acquired pr-1-like genes via horizontal gene transfer from the host plants during the coevolutionary arms race.” – I know speculation is acceptable in the discussion but I am not sure this is supported by the literature since most organisms contain a CAP-domain containing protein (or CMB motif) including human glioma PR-1 protein and a snake-venom cysteine-rich secretory protein, yeast as you described. This is also unlikely since the plant PR-1’s don’t contain the Ser/Thr rich region whereas PRY1 does. I think it would be better to rephrase this section.
- Line 326-328 – “PR-1La lacks a transmembrane domain, it is not clear how the signals perceived at the cell surface is transduced to activate the downstream signaling pathway.” Suggest edit it to “PR-1La lacks a transmembrane domain, therefore it is not clear how the cell surface signals would be perceived and transduced to activate downstream signaling.”
- Line 387 – “performed as described” – incomplete sentence!

REVIEWER COMMENTS

Reviewer #1 (Remarks to the Author):

PR-1-like proteins belong to a large protein family, and they have been implicated in many different biological processes including immune defense and cancer progression. These proteins share a conserved CAP domain and are mostly secreted glycoproteins. Their mode of action, however, remains poorly characterized. Thus, progress in understanding the function of these proteins at a molecular level would be a welcome advancement.

Lin et al. now report that the corn smut *Ustilago maydis* pathogenesis-related 1-like protein (UmPR-1La) acts in detoxification of phenolic compounds, and that its Ser/Thr-rich N-terminal domain mediates cell surface association. In addition, the authors report that UmPR-1La has gained a specialized activity in eliciting hyphal formation and they suggest that the protein senses phenolic compounds to direct hyphal growth in plants. The third reported finding is that UmPR-1La is cleaved by a plant cathepsin B-like protease to release a functional signaling peptide (CAPE-like) that suppresses plant immunity and promotes fungal virulence.

The main limitation of this study is that it presents an assembly of three more or less independent observations but fails to convey a compelling and significant overall progress in our understanding of the mode of action of PR-1-like proteins. Moreover, each of the three findings is not sufficiently supported by data.

We thank the reviewer for the valuable comments and apologize for the incomplete conveyance of the findings in the previous version of the manuscript. To rectify this, we have made necessary modifications to the respective sections of the results and discussion, and have introduced a new model to effectively communicate the findings of this study. Apart from this, in response to the concerns raised by the reviewer, we have carried out additional experiments to specifically address those issues.

Specific comments:

1). Fig1a, the consensus sequence for the CAPE1 cleavage is CNYx. It is not clear why the authors change this consensus to CxYx. The same applies for the 11-amino acid CAPE-like peptide sequence. The plant PR-1 derived CAPE1 peptides all end in R/KPY but this is not conserved in UmPR-1L. It is not clear what criteria these authors apply to define the CAPE-like peptide derived from UmPR-1La as being similar/like to that of the plant CAPE1 peptides.

We thank the reviewer for bringing these mistakes to our attention. We have made the necessary corrections in the figure legend of Fig.1a. Regarding your second question about the definition of the CAPE-like peptide (CAPE-L) from UmPR-1La, we compared the peptide sequences between smut fungal PR-1Ls and plant PR-1s (Fig. 1b), and revealed a high conservation of the first seven amino acids after CNYx motifs of PR-1 and PR-1L proteins. Consequently, we have designated peptides containing the MEME-derived consensus sequence P[P/V]GN[Φ/V][I/V]G as CAPE-like peptides (CAPE-L). The peptide sequence comparison has been incorporated into Fig. 1b in the revised manuscript. Our current hypothesis is that UmCAPE-La may compete for binding to the plant CAPE receptor, thereby blocking the CAPE-dependent signaling pathway. This hypothesis could be tested in the future once the receptor is identified.

The green line denotes the consensus sequence conserved in fungal PR-1Ls and plant PR-1s.

2). Fig1b, the normalized read counts for pr-1a appear to be very low, especially when compared to those for pr-1b, which are 100-fold higher (FigS1b), or CatB3, which are about 500-fold higher (Fig5a). The gene thus appears to be expressed only at very low levels.

We have observed this discrepancy, and one possible explanation is that *PR-1La* may have longer mRNA and protein half-lives or exhibit higher translation activity. However, it is important to note that this hypothesis remains speculative. In our recent study, we investigated the expression of the *Scdda2* gene, which was found to be 200 times more abundant than the *Scdda1* gene. Surprisingly, despite this significant difference, only *ScCda1* proteins were detectable (<https://doi.org/10.1128/mbio.00093-23>). This finding suggests that there may not always be a positive correlation between gene expression and protein levels. Moreover, a high expression level of effector genes might not necessarily correlate with their contribution to the virulence of *U. maydis*.

3). Fig1d, are the disease symptoms between the single mutant (Δa) and the double mutant ($\Delta\Delta ab$) statistically significant? The authors describe these differences as “more pronounced” (L137).

After re-analyzing the data obtained from the four biological replicates, there is no statistical significance in disease symptoms caused by these two mutants. As a result, we have revised our statement to read as ‘a similar decrease in virulence was also observed in the double mutant $\Delta\Delta pr-1lab$ ’.

4). Fig2a,b, the authors say that UmPR1-La and ScPry1-mCherry are cell surface associated proteins, and that UmPR1-La localizes to bud necks and growing tips in sporidial cells. However, they do not show any colocalization experiments with established cell wall markers. At the same time, they show that these

proteins are secreted, but they do not include a cell wall-associated protein in the Western blots as control. Are these proteins now secreted or are they cell wall associated?

To confirm the localization of UmPR-1La on the cell surface, we conducted an experiment involving the overexpression of HA-tagged UmPR-1La in the AB33 strain. The AB33 strain was chosen for its efficient filament induction, controlled by the nitrate-inducible *nar1* promoter, when cultured in a nitrate-containing liquid medium, enabling us to perform plasmolysis to separate the cell wall and plasma membrane. Immunostaining was performed, along with labeling of chitin using WGA-AF488. Subsequently, plasmolysis was carried out. The results demonstrated the colocalization of UmPR-1La with the chitin on the cell surface, while the fluorescence of UmPR-1La was not detected inside the expanded space resulting from plasmolysis (*). This provides conclusive evidence for the cell-wall localization of UmPR-1La. The corresponding data has been incorporated into Fig. 2c.

Regarding the concern about protein secretion, it is important to note that, due to the overexpression conditions and the absence of a GPI-anchor, certain cell-wall localized proteins may not bind tightly to the cell walls, as GPI-anchored proteins do. Consequently, these proteins can be detected both on the cell walls and in the culture supernatant if overexpressed. Our finding is also in line with other cell-wall protein studies (Tanaka et al., 2020, <https://doi.org/10.1111/nph.16508>; Cottier et al., 2020, <https://doi.org/10.1242/bio.053470>).

5). Fig2e, shows that recombinant full length and the Ser/Thr-rich domain of UmPR1-La bind chitin and chitosan. The authors do not show whether these proteins are soluble under these conditions in the absence of chitin or chitosan. If the Ser/Thr-rich domain would promote protein self-association, it would appear as binding to chitin/chitosan. These proteins were expressed and purified from bacteria, nevertheless, the authors suggest that they associate with the fungal cell wall by binding to chitin/chitosan via glycosidic linkages (L169-172).

We appreciate the concern raised by reviewer #1 regarding protein solubility, which we initially overlooked. As suggested, we conducted protein solubility tests in the assay buffer (50 mM Tris-Cl and 150 mM NaCl, pH 8) without ligands. However, the results were inconsistent in the three replicates (shown below). Full-length (FL) and S/T recombinant proteins showed a tendency to be insoluble, while CAP-domain proteins appeared to be more soluble. Due to these inconsistent findings, we have decided to remove the results of the polysaccharide binding assay from the revised manuscript.

Does the CAP domain bind lignin and is this abrogated in a CBM (FY*) mutant version? This seems to be important given that the authors subsequently show that the CAP domain binds phenolic compounds that are the precursors and building blocks of lignin.

To perform the lignin binding assay, we tested the solubility of lignin in the buffer (50 mM Tris-Cl and 150 mM NaCl, pH 8), and observed that the buffer turned brown, indicating that lignin might dissolve in the buffer. Under this condition, all the recombinant proteins were detected in the supernatant fractions in the presence of lignin. However, this result does not indicate that the FL or CAP domain proteins do not bind phenolics. They could bind to the phenolics released from the lignin in the supernatants.

We next attempted the binding assay in H₂O, where lignin is insoluble. However, the solubility of FL recombinant proteins in H₂O without lignin was inconsistent, even though they appeared in the supernatant fractions with lignin. Similarly, the solubility of S/T or CAP-domain proteins was inconsistent, with CAP proteins tending to be found more in the pellet fraction with lignin. Given that lignin is a complex, three-dimensional highly branched polymer and whether CAP-domains are able to access phenolics in such a complex or only bind to free phenolics is unclear, and the inconsistent solubility test results prevent us from reaching a definitive conclusion, we therefore report our findings here for the reviewers.

6). Fig3, the apparent role of UmPR1-La in conferring eugenol resistance and the putative function of UmPR1-La in sensing phenolic compounds to signal pseudohyphal growth is based on cells that overexpress this protein, i.e., a SG200 wild-type strain expressing an additional copy of UmPR1-La from a constitutive promoter. Thus, this is a gain-of-function phenotype that may have little in common with the native function of this gene/protein. These crucial experiments need to be supported by showing that mutant cells lacking one (Δa) or both ($\Delta\Delta ab$) UmPR1-L are hypersensitive to eugenol, and that this sensitivity and hyphal growth can be complemented in the deletion strain by expression of one or both UmPR1-L from their respective native promoters. The putative function of the CBM should then be tested in this sensitized background and not in an overexpressing strain. Are the WT and FY* mutant proteins used in Fig3c and Fig4b expressed at comparable levels, please provide Western data.

We express our gratitude to the reviewer for this comment, which indicates that our description in the text was insufficient. The gene expression data shown in Fig.1d indicates that *U. maydis* cells grown in liquid medium do not express *PR-1La* and *PR-1Lb* genes, and induces their expression during the biotrophic development. As a result, the wild-type strain SG200 is hypersensitive to eugenol due to the absence of native *PR-1La* gene expression. Therefore, the protection of cells against eugenol requires the overexpression of PR-1La proteins in SG200. To ensure readers are reminded of this fact, we have added the sentence "*PR-1La* and *PR-1Lb* genes were not expressed in axenic culture, but they were induced during biotrophic development of *U. maydis* SG200."

With regards to protein levels of PR-1La and PR-1La(FY*) expressed in SG200, we have now included the protein expression and secretion data in the revised manuscript to show that the comparable levels of the proteins expressed in the cells (Fig. 3c).

7). Fig4, as in Fig3, the authors should make sure that they are not scoring an overexpression phenotype. Is an SG200 strain not expressing WT or FY* mutant version of UmPR1-La sensitive to ferulic acid, what is its morphology? The binding assay shown in Fig4c does not indicate whether binding is specific, concentration-dependent, and saturable. Could it be that the protein gets denatured in the presence of ferulic acid and that this is scored as "binding"? This also applies to FigS4b.

In Fig. 4b, we have included microscopy data depicting the cell morphologies of SG200, as well as the overexpression strains SG200_PR-1La and SG200_PR-1La (FY*), with and without ferulic acid (FA) treatment. After exposed to FA, both the SG200 and SG200_PR-1La (FY*) strains displayed similar morphology and appeared slightly less healthy in the presence of FA. However, the presence of a hyphal-like structure was exclusively observed in the FA-treated SG200_PR-1La strain.

Regarding the FA binding assay, our titration results demonstrated the dose-dependent binding of PR-1La protein to FA. We observed that the saturation of FA binding occurred before reaching a concentration of 10 μM FA, as evidenced by the presence of a peak in the absorption spectrum of free FA at approximately 420 nm. In contrast, the mutant protein PR-1La(FY*) exhibited no response to different FA concentrations. The fluorescence of FA at 330 nm was detected but remained constant regardless of the FA concentration. These findings are now presented in Supplementary Figure S4b.

We thank the reviewer for raising the concern regarding protein stability in the presence of phenolics, an aspect we had overlooked. To address this concern, we investigated the stability of the protein after exposure to phenolics for 15 minutes prior to fluorescence measurement. Our observations revealed no substantial protein degradation following the incubation period, suggesting that the observed fluorescence change is indeed attributed to the binding of phenolics to PR-1La. We have included one of the replicate experiments in the revised manuscript as Supplementary Figure S4d.

8). Fig5, the authors fail to detect the UmCAPE-La peptide in the apoplastic fluid of *U. maydis* infected leaves (L239). Hence there is no evidence that this peptide is actually produced in planta. Did they try infection with an overexpression strain? Fig5b, CatB3, which shows cleavage activity towards recombinant UmPR1-La, was purified from the apoplastic fluid of tobacco leaves and this protein preparation is not pure (FigS5b). Purification and cleavage assay with a catalytically dead mutant version of CatB3 would be needed to claim that the observed cleavage is indeed due to CatB3 and not a contaminating protease. This seems to be particularly important given that the cleavage of the native secreted version of UmPR1-L by CatB3 appears to be very slow and inefficient.

- We attempted infections using various overexpression strains, including those with the *otef* promoter or multiple copy integration under the control of the native promoter. Additionally, we collected the apoplastic fluids at 1 dpi and 8 dpi, but we were unable to successfully identify the UmCAPE-La peptide. We lack a satisfactory explanation for this outcome but hypothesize that the UmCAPE-La peptide's concentration might be low, or it could potentially bind to an unidentified receptor upon being released.
- We acknowledge that there is a low efficiency of cleavage by CatB3 in our assay under the current conditions. We do not have a definitive explanation for this observation. However, we speculate that either an unknown factor is required for the activity of CatB3, or CatB3 may not be able to efficiently access the CNYD motif for cleavage.
- In response to the concern raised by reviewer #1 about the purity of CatB3 proteins, we fully agree on the importance of having an inactive form of CatB3. To address this, we introduced mutations in the catalytic site of CatB3 (Cys121 and His276). Both the wild-type and mutant versions of CatB3 were purified in parallel, and the protein purity is illustrated in Fig. S5e. We found that both the catalytic inactive mutants (C121A or C121A H276A) blocked the cleavage of UmPR-1La, while the wild-type CatB3 consistently cleaved UmPR-1La. These results provide support for our claim regarding the cleavage by CatB3 (Fig. 5c).

9). Fig6, if CAPE-L indeed has immune suppressive activity one would expect that *U. maydis* strains overexpressing UmPR1-La show enhanced virulence. What is the induction of PR gene expression when non-infected plants are treated with ZmCAPE, CAPE-L, or both peptides together, compared to plants treated with scrambled peptide sequences?

Overexpression of UmPR-1La might result an increase in *U. maydis* virulence. However, this is not always happened, particularly in *U. maydis*. We had overexpressed it using the promoter of *UmPR-1Lb*, and it did not complement the virulence of the *pr-1la* deletion mutant. Nevertheless, we followed the suggestions by infiltrating peptides into maize leaves, followed a protocol as described in a previous study (<https://doi.org/10.1038/s41477-018-0116-y>). Briefly, the synthetic peptides were dissolved in water, and the peptides was infiltrated using a blunt needled syringe at the base of fourth leaves of 10-day old plants. The leaf samples were harvest 24hr later. We attempted using both 0.6 μM and 2 μM peptides for infiltration and did not observe any significant differences among the treatments. Due to the structure of maize leaves, it was challenging to avoid causing wounds during peptide inoculations. The variations in *PR* gene expression may be attributed to these wounds. Consequently, we present these findings to the reviewers. Presented below are the results obtained from three (2 μM) and five (0.6 μM) independent biological replicates.

Mixed peptides: Mixed with an equal concentration of CAPE-La and ZmCAPE.

Scrambled peptides: contains an identical amino acid composition to CAPE-La.

Maize *GAPDH* gene was used for normalization. The *PR* gene expression level in H₂O was set to 1.0. Average values of three or five biological replicates are shown. Error bars indicate \pm SD

Despite our inability to demonstrate the role of ZmCAPE in plant defense through peptide infiltration, we successfully exhibited a notable reduction in the virulence of *U. maydis*. This outcome was achieved by substituting UmCAPE-La peptide of UmPR-1La with ZmCAPE, and delivered them via *U. maydis*. (Figure 6e-f). (Please refer to the response to question #13).

Reviewer #2 (Remarks to the Author):

The authors provide a functional characterization of 1 (2) effector(s) of the fungal maize pathogen *Ustilago maydis*, which have homology with plant PR1 proteins, including the C-terminal CAPE peptide which is known to play an important role in PR-1 mediated induction of plant defenses. The findings shown in the paper are new and exciting. Also, the experiments are all over sound. However, the presentation and documentation of data is partially incomplete. Also, this reviewer is not completely convinced how the two proposed functions of eugenol detoxification (which is actually rather resistance/tolerance than detox) and CAPE-release to block plant immunity are linked. Also, it appears unclear why/how the Um-PR1 released CAPE interferes with the generation of Zm-derived CAPE peptide.

General questions and comments:

9). Are orthologs of related smuts able to complement the PR-1La disease phenotype, or is this specific to the *Ustilago maydis* protein?

We appreciate the review for raising this question, as it has provided us with an opportunity to uncover additional interesting findings for further exploration in the future. In response to this question, we generated a complementation strain using the ortholog Sr10279 from *Sporisorium reilianum*, a maize-infecting fungus. Despite Sr10279 being able to localize to the *U. maydis* cell wall and possessing the conserved CNYD and CBM motifs, it was unable to rescue the virulence of *U. maydis* $\Delta pr-1la$ strain, and it did not form pseudohyphae upon exposure to ferulic acids. This discrepancy may be attributed to potential divergence in downstream signal reception partners between Sr10279 and UmPR-1La. Consequently, Sr10279 is incapable of substituting the role of UmPR-1La in eliciting hyphal-like structures in *U. maydis*. The corresponding data has been incorporated into Fig 4c-e.

10). Does the PR-1La deletion mutant have any (growth/developmental) defects besides in-plant development? Given its function at cell-surface localization this seems to be relevant for the interpretation of the results and the actual biological function of the PR-1La.

The deletion mutants, complementation strain ($\Delta pr-1la_PR-1La$; controlled by the native promoter), and the overexpression strains (SG200_ $PR-1La$ and SG200_ $PR-1La(FY^*)$) driven by the *otef* promoter) displayed no visible abnormalities during their growth in a liquid medium or in their ability to form filaments on charcoal plates. Their morphologies were comparable to SG200. As a result, we can conclude that $PR-1La$ is associated with the *in-planta* development of *U. maydis*. These results have been included in supplementary figure S1d.

11). Based on the expression profiles, both *pr-1* genes are not expressed when cell growth in liquid culture.

- It appears (particular at level of this journal) somehow inappropriate to show gene expression data in the main figures (Fig 1; Fig 5) that is actually just extracted from a publication by others. Authors may provide own qPCR data to confirm the proposed expression patterns in their own experimental conditions (and refer to the published data in the supplements).

We completely agree with the reviewer. We have removed the expression data and conducted a qRT-PCR assay to analyze the expressions of *UmPR-1La* and *UmPR-1Lb* using samples from SG200-infected leaves. These figures have been added to the revised manuscript (Fig. 1d and S1b).

12). The figure labels are missing some relevant information to make them accessible. In Fig 2 it took me a while to understand that I am looking on an “*otef*” OE which explains the signals in sporidia. In Figure 3C I needed to search for “*FY**” and found it in the legend for 3D. However, as the figure 3C is shown, one would understand that there is a “*WT*” and a “*FY**” both being somehow with “*SG200_PR-1LaHA*”. What I guess it is, is strain SG200 (which actually should not be called a “wild type”, because it is an artificial lab strain) vs the *pr-1la* deletion strain. Right?

Allover, it took me lots of time to look up this kind information up, as descriptions in the figures are sparse/sometimes misleading and also the legends lack information (e.g. legend Fig 3 “same strains used in 2a” – this is not sufficient. Please name the strains which are shown!) Thus, the authors should carefully revise their figures (including supplements, where also info is missing), provide more precise labels and information on strains wherever it is needed, otherwise it is very difficult to follow and interpret results.

We offer our sincerest apologies for the inadequate and inattentive explanation provided, as well as the incorrect labeling of the experimental data depicted in the figures. To avoid any possible confusion, we have made the necessary revisions, substituting 'WT and FY*' with 'PR-1La and PR-1La(FY*)' in both the text and figures. Additionally, we have made every effort to provide comprehensive information in the figure legends within the given word limits.

13). Does the CAPE-peptide have a virulence function for *U. maydis*? Figure 6 implies that it has a role in virulence. So, is a mutant with a C-terminally truncated PR-1La reduced in virulence? If so, can other (e.g. Zm) CAPE sequences not complement this?

We appreciate the reviewer for suggesting this experiment, which allowed us to confirm the role of ZmCAPE in priming maize defense. Following the suggestions of the reviewer, we generated the complementation strains in which the UmPR-1La's CAPE-La sequence was replaced with either ZmCAPE or UmCAPE-Lb after the CNYD motif. The complemented strain expressing PR-1La (CAPE-Lb) proteins under the PR-1La promoter partially restored the virulence of $\Delta pr-1la$, resulting in an intermediate phenotype that showed no significant difference from SG200 or $\Delta pr-1la$. Surprisingly, the virulence of the complementation strain $\Delta pr-1la_PR-1La(ZmCAPE)$ was severely affected, leading to the development of tiny tumors with a size of less than 1 mm. The significant decrease in disease symptoms caused by ZmCAPE suggests a role of plant defense, and the spatial and temporal delivery of ZmCAPE peptides by *U. maydis* could effectively boost plant immunity to suppress fungal virulence. This negative impact could be partially reversed by the application of UmCAPE-La peptides, further supporting the contradictory role of UmCAPE-La and ZmCAPE in regulating plant immunity during the maize-*U. maydis* interactions. We have now included the data in Figure 6d-e.

14). If *Ustilago* CAPE competes for Zm-CAPE with the receptor – how is it then not activating it (if it has a stronger binding affinity?)

Based on our findings in Fig 6e, it appears that the use of UmCAPE-La could mitigate the negative impact of ZmCAPE on *U. maydis* virulence. Our current hypothesis is that UmCAPE-La may compete for the unidentified receptor binding, leading to the inhibition of the receptor's role in activating PR gene

expression (as depicted in Fig. 6f). In maize B73, among the seventeen PR-1 genes examined, four contain peptides longer than 11 amino acids that do not end with [R/K]PY, while the remaining peptides end with [R/K/S]P[Y/F] (Am J Transl Res. 2022; 14(11): 8315–8331). We speculate that the first seven conserved amino acids found in UmCAPE-La and maize CAPE peptides could be sufficient for receptor binding, while the last few amino acids of the peptides might influence the degree of receptor activation. The precise mechanism by which plant CAPE triggers the SA-signaling pathway remains unclear. However, once the receptor is identified, further investigation can be conducted to assess the role of these amino acids in modulating plant CAPE-dependent signaling pathway.

Specific questions:

15). In L136 it is stated that “double deletion phenotype is more pronounced” – Is this really significant compared to the single deletion?

The same argument is also raised by Reviewer#1. We have re-analyzed the data and there is no significant difference between single and double deletion mutants, and we have reworded it to ‘a similar decrease in virulence was also observed in the double mutant $\Delta\Delta pr-1lab$ ’.

16). Figure 3 and corresponding text: what is the evidence for a “detoxification” – what is shown is resistance to a toxic compound, but not its detox.

We fully concur with the reviewer's assessment, as we did not have evidence to substantiate its detoxification properties. In the revised manuscript, we have now replaced it with the term 'resistance'.

17). Figure 4C –what is actually detected here by the spectrophotometer? At which wavelength? Why are no biological reps shown but only technical reps?

Fluorescence spectrophotometry could be used to study the ligand binding-induced conformational change of a protein. The natural fluorescence of tryptophan is used as a reporter of the conformational changes induced by the ligand binding. Tryptophan were excited at 290 nm, and emission spectra were recorded in the range of 310–450 nm. The experimental detail has been provided in ‘Methods’ section. In our previous manuscript, we chose to present only one biological replicate because the figure with three biological replicates appeared crowded and untidy. The excessive number of data points resulted in extensive error bars, leading to visual clutter. Due to the concern regarding the absence of biological replicates, we have now replaced it with a new one that displays the mean and standard deviation of all three biological replicates (Fig. 4f).

18). - Figure 5 – also here, biological reps are not shown (MS data). Also, it is not possible for me to understand the figure completely. What are the sequences below 5b? What is shown in 5C (what is “y7” “y6” etc?) What is scale of “intensity”? How does this show a specific cleavage of PR-1La by CatB3?

We apologize for not providing sufficient information to access Figure 5e. We have revised the figure legend for clarity. The peak area of fragment ions was used to quantify the quantity of tryptic UmCAPE-La peptide in both the -CatB3 and +CatB3 samples. Fragment ions y6 and y7 extend from the C-terminus, and y7++ is a doubly charged ion. The b5 ions extend from the N-terminus. To illustrate the fragment ions of the peptide, we have included a cartoon in Figure 5d. The term “intensity” refers to the amplitude of the free induction decay signal. Because the intensity is not a precise measurement to represent the absolute peptide abundance, therefore it is usually not labeled.

Regarding the issue of biological replicates, the His- and HA-tagged UmPR-1La served as two separate replicates for CatB3 cleavage and consistently produced similar results. This observation indicates that only the +CatB3 sample displayed fragment ions derived from the UmCAPE-La peptide.

19). - Fig S5A indicates degradation of PR-1La by apoplastic fluid of maize. However, none of the inhibitors used could fully block this degradation completely. This actually suggests that multiple proteases are involved in degradation of the protein beyond Cat3B. How do the authors interpret this?

Thanks for the question. We believe that the insufficient amount of protease inhibitors added is the reason for this. To address this issue, we conducted a titration assay by adding varying amounts of DMSO-dissolved E-64 to apoplastic fluid proteins (AF). In this assay, we also reduced the amount of AF by half, using 15 µg AF instead of the amount used previously in Fig.S5A. Our findings indicate that the cleavage of PR-1La in the apoplast was blocked in the presence of E-64. Additionally, E64 also inhibited the activity of CatB3. These results suggest that the cleavage of UmPR-1La in the apoplast is caused by cysteine proteases, and CatB3 is one of them. These figures have been incorporated into Fig. 5a and 5b of the revised manuscript.

Reviewer #3 (Remarks to the Author):

The submission, “Ustilago maydis PR-1-like protein has evolved two distinct domains for dual virulence activities” by Lin et al describes a fungal protein, PR-1-like, that has homology to the well described yeast, PRY1 and plant PR1. The authors show that the U. maydis PR-1-like proteins contains two domains, Ser/Thr-rich region and the CAP domain with a predicted C-terminal peptide similar to PRY1, while plant PR-1’s only contain the CAP domain with a C-terminal peptide. The authors show that the Ser/Thr-rich region is responsible for the binding of PR-1-like to Chitin and therefore its localisation to the fungal cell wall. While the CAP domain is responsible for sterol/phenolic binding which induces a switch in U. maydis from sporidia to pseudohyphal structures which allows survival of the fungi. Additionally, the authors show that the C-terminal peptide of PR-1-like is cleaved by CatB3 a member of the cysteine protease family and that PR-1-like can modulate known plant immune genes which is likely to occur via the C-terminal peptide. Overall, this paper is quite well written and logically structured making it relatively easy to follow the narrative, however some work on Figure layout and labelling will help greatly. I have below some major and minor points to address which should improve the article prior to publication.

Major point to address

20). Figure 2 – These microscope images (a and d) are not good enough quality to be the only evidence of cell surface localisation. I would like to see better quality and higher resolution images as well as co-localisation with a known cell surface protein so show co-localisation. Arrows would also help to direct reader.

Same question (#4) regarding the cell-wall localization is also raised by reviewer #1. To address this issue, we performed co-localization experiments of UmPR-1La with chitins in the AB33_PR-1La strain treated with plasmolysis, and found that UmPR-1La co-localized with chitin on the filamentous cell surface and not located inside the space enlarged by plasmolysis. Please refer to our response to question #4 raised by reviewer #1.

To improve the image resolution in Fig. 2, we have replaced them with high-resolution enlarged images. This clearly shows the localization of PR-1-like proteins on the cell surface.

21). Please review all figures as many are lacking good labelling e.g. Fig2, no labels on western blot for expected size of protein especially important for PR-1Lb as there are multiple bands and the band sizes don’t seem to match my calculations based on the amino acids described in fig 1. Also Fig2a - PRY1HA I believe should be labelled Umsp:PRY1-HA this was very unclear as the labelling did not match the description in the text of the results. These are just two examples.

We apologize for any confusion caused by our previous lack of clarity in the figure legends. We have now taken the necessary steps to revise all figures and their corresponding legends to ensure that they are clear and unambiguous to readers. Thank you for bringing this to our attention, and we hope that our revised figures and legends will be more helpful to readers.

We have noticed that the molecular weights of PR-1-like proteins are higher than anticipated. Specifically, PRY1 expressed in SG200 cells exhibited slower migration than expected in SDS-PAGE, despite an expected weight of 33kDa, it was detected at 72kDa. This finding is consistent with the result from Roger Schneider's team, who discovered that yeast PRY1 is a high molecular weight glycoprotein that migrates at >70kDa in SDS-PAGE (<https://doi.org/10.1242/bio.053470>). Based on this information, we suspect that PR-1L proteins are also glycosylated to some extent.

22). Add in the O-glycosylation data into a sup figure

Due to the inconsistency in protein solubility, we have excluded the polysaccharide binding result from Figure 2e (Please refer to our response to question #5).

23). Figure 3b – Given that the authors base this experiment off the PRY1 literature of PR1 binding Eugenol this seems an unexpected result that the cells do not survive, please expand on/discuss this. Could it be linked to the data in Fig2a where PRY1 does not localise to cell surface in *U. maydis*? Make clear your reasons for reader.

Based on this question, we have realized the problem of providing insufficient explanation in our previous manuscript. We have made efforts to address this issue. According to our findings, PRY1's inability to protect *U. maydis* cells is not solely due to localization and eugenol-binding issues. This is evident from the fact that the chimera protein, despite being localized to the cell wall, failed to provide protection. Additionally, it has been reported that the CAP domain of PRY1 can bind eugenols. The absence of hyphal-like structures is the key reason why PRY1 fails to protect *U. maydis* cells but not *S. cerevisiae* cells (Fig. S3 and Fig 3b). These hyphal-like structures are essential for protection in *U. maydis*, whereas complete shielding of *S. cerevisiae* cells by PRY1 is sufficient to prevent eugenol toxicity.

To induce UmPR-1La-shielded hyphae, UmPR-1La needs to transmit signals to its specific receptor, which likely does not interact with either PRY1 or Sr10279, leading to the failure in inducing hyphal-like structures in *U. maydis*. However, we speculate that Sr10279, based on conserved motifs and its grouping with UmPR-1La in the same clade (Fig 1b-c), can induce such structures in *S. reilianum*. We have discussed this aspect in the line 251-256 in the results section.

- Add loading controls to immunoblot panels.

We have now provided necessary loading controls to immunoblots.

24). I find it surprising not to have seen an enhancement of defence by the application of the maize CAPE peptide. I would suggest you repeat this experiment, but this time pre-treat the maize with ZmCAPE 1 day prior to infection. This experimental set up is how previous publication have conducted the experiment e.g. Chen et al, 2014 Plant cell and Sung et al, 2021 New Phyt (refs 8 & 9 on your list). Have you also tried it with WT SG200?

We had conducted this experiment previously, where we pre-treated maize seedlings with ZmCAPE a day before *U. maydis* infections. However, we did not observe any reduction in the virulence of SG200 or $\Delta pr-1lab$, which was puzzling to us as we lacked a satisfactory explanation. In this revision, we followed the suggestion of reviewer#2 (please refer to our response to Question #13) by replacing the UmPR-1La's UmCAPE-La sequence with ZmCAPE and delivering it via *U. maydis*. Using this delivery method, we observed a significant decrease in the disease symptoms of this complementation strain. This finding supports the conserved role of ZmCAPE in priming plant immunity and also suggests that the lack of enhanced maize resistance by the application of ZmCAPE could be due to the timing and method of peptide delivery in maize. Furthermore, the negative impact of ZmCAPE on *U. maydis* virulence delivered by this strain was mitigated by the application of UmCAPE-La peptides. These figures have been included in the revised manuscript (Fig. 6d-e).

Minor point to address

- Line 79-81 “However, it is unclear whether a protease involves in the cleavage of the CNYx motif and a receptor to perceive CAPE peptide to activate” suggest editing to read “However, it is unclear whether a protease is involved in the cleavage of the CNYx motif and if there is a receptor to perceive CAPE

peptides that activate”

We have replaced the sentence as suggested.

- Line 95 – “sexual life cycle, it requires *U. maydis* fine sensing and integrating environmental” suggest editing it to “sexual life cycle, *U. maydis* requires fine sensing of environmental”

The sentence has been modified.

- Line 97 – “*U. maydis* secrete a” edit to “*U. maydis* secretes a”

It has been edited.

- Line 108 – “elicit hyphal-like structures. A CAPE-like peptide” suggest edit to “elicit hyphal-like structures, while a CAPE-like peptide”

This sentence has been modified.

- Line 172-173 – I think this statement has not been 100% proven therefore I would say “the results indicates that this PR-1La may incorporate into the cell-wall chitin/chitosan-glucan matrix possibly via glycosidic linkages”

The insolubility issue of recombinant proteins in the absence of a ligand prevented the confirmation of the finding that PR-1La protein binds to chitin/chitosan in the polysaccharide precipitation assay. This statement has been deleted (Please refer to our response to question #5).

- Remind reader that PR-1-likes are not expressed in culture therefore the native PR-1-like’s are not affecting the phenolic binding assays.

Thanks for your suggestion. A sentence has been included at the start of the text as a reminder for readers that PR1-L proteins are not expressed when cells are grown in liquid culture.

- Figure S3a – there is wrong labelling between the graph and the images or there is a wrong image added – graph shows PR-1La showing cell survival and growth however image is showing PR-1Lb data. Need consistency between graph and images, suggest adding PR-1La and PR-1Lb to both.

To prevent unnecessary duplication and ambiguity, we have followed your suggestion and rearranged the images and placed in fig. 3b.

- Line 193 – Wrong Figure, should read Fig 2a.

We have made the corrections.

- Fig3d – suggesting adding in WT complementation line. Also inconsistent labelling between Fig 1d and 3d

To have consistency in labeling and prevent any confusion, we have relabeled the strains that express wild-type and mutant PR-1La proteins as SG200_PR-1La and SG200_PR-1La(FY*), respectively.

- Line 218 – CA-treated PR-1La (FY*) cells show a swollen morphology, based on the image shown for PR-1La WT treated with CA these pseudohyphae also look swollen.

U. maydis exhibited greater sensitivity to coniferyl-alcohol (CA) than to ferulic acids and coumaric acids. Since we did not titrate the dosage of CA to determine the minimal effective concentration for inducing pseudohyphae, the use of 5 mM CA in our treatment assay is likely excessive, resulting in damage to the wild-type PR-1La protein-expressing cells.

- Inconsistent labelling of sterol graphs between Fig3 and 4. FigS3 and FigS4

We have now consistently re-labeled all graphs in the revised manuscript.

- Lines 241-244 – re-word sentence, it is difficult to follow. You need to get the phrase culture filtrate or culture supernatant in there as well as saying U. maydis SG200 +PR-1La-HA line.

We have rephrased the sentences to “we explored UmPR-1La cleavage by incubating the culture supernatant of SG200_PR-1La strain with the apoplastic fluid of SA-inoculated maize leaves, along with different protease inhibitors”.

- Figure S5a legend is missing information – treatments and concentrations used. Labelling of figure and description in materials and methods don’t match i.e. is control on figure the PIC as in materials you mention DMSO is the control?

We have added the detailed information in figure legend of fig. S5a.

- Line 249 – “Cathepsin B-like 3 (CatB3) has a similar expression profile as umpr-1la” I disagree PR-1La expression is all over the place. I would just say CatB3 is induced during infection or CatB3 is strongly induced from 4dpi compared to mock treatment.

We have modified the sentence to “*Cathepsin B-like 3 (CatB3)* is expressed in the early stage and strongly induced from 4dpi and onwards”.

- Fig5b add arrow to indicate PR-1La band also labels for Coomassie and immunoblot.

The previous fig. 5b is now moved to fig. 5d. Black and red open arrows indicate full-length and truncated PR-1La proteins respectively, have been added in both Coomassie-blue-staining PAGE and immunoblot.

- Fig5b make the second blot on U. maydis secreted PR-1LaHA blot Fig 5c as currently the legend is hard to follow what is being described. Quantify band intensity against total protein stain as there is no size shift (due to small cleavage size) as observed in the previous gel and no loading controls shown.

The immunoblot for UmPR-1LaHA in the previous Figure 5b has been replaced with a new figure, which includes the catalytically inactive mutants of CatB3 to support the cleavage of UmPR-1La by CatB3 (Figure 5c). In this new figure, we have also included a silver-stained gel to serve as a loading control.

- Line 265 – “In maize, homolog Xcp2 is highly expressed in the early infection process and then slowly declines during biotrophy (Fig. S5d).” – I disagree with this statement, there is no difference in expression of Xcp2 between mock and infected tissue. Remove or reword.

The sentence has been rewritten as ‘*Xcp2* is expressed at all infection/wounding stages.’

- Line 268 – suggested edit “truncated fragments were detected by Coomassie-blue-staining-PAGE and immunoblot for PRB1-3 and PR-1b”

The sentence has been edited accordingly.

- Line 313-317 – “U. maydis probably acquired pr-1-like genes via horizontal gene transfer from the host plants during the coevolutionary arms race.” – I know speculation is acceptable in the discussion but I am not sure this is supported by the literature since most organisms contain a CAP-domain containing protein (or CMB motif) including human glioma PR-1 protein and a snake-venom cysteine-rich secretory

protein, yeast as you described. This is also unlikely since the plant PR-1's don't contain the Ser/Thr rich region whereas PRY1 does. I think it would be better to rephrase this section.

Thanks for pointing it out. In a study on the genome-wide analysis of maize PR-1s, it reported that three out of the 17 maize PR-1s contain an extension region at their N-termini. This extension region is notably enriched in Ser, Thr, and Gly (Am J Transl Res 2022;14(11):8315-8331). However, it is unknown how fungal pathogens acquire *PR-1L* genes during co-evolution. We agree that the previous speculation has no support by literature, we have rephrased the sentences to ' While the exact origins of *PR-1L* genes in fungal pathogens remain unclear, their capacity to act as substrates for plant proteases and conservation at CAP domain residues underscore a related role in plant-pathogen interactions during the process of co-evolution'.

In addition, we also added the sentence 'the extension region is also present in three out of the 17 maize PR-1 proteins' at the beginning of the Result section.

- Line 326-328 – “PR-1La lacks a transmembrane domain, it is not clear how the signals perceived at the cell surface is transduced to activate the downstream signaling pathway.” Suggest edit it to “PR-1La lacks a transmembrane domain, therefore it is not clear how the cell surface signals would be perceived and transduced to activate downstream signaling.”

The sentence has been edited as suggested.

- Line 387 – “performed as described” – incomplete sentence!

The sentence has now been completed and written as ‘The transformation of *U. maydis* and genomic DNA isolation procedures were performed as described in a previous study’.

REVIEWER COMMENTS

Reviewer #1 (Remarks to the Author):

We appreciate the author's efforts to address our concerns and think that the revised version of this manuscript has been greatly improved. Our main concern, however, that the results do not provide a strong coherent picture of the function of UmPR-1La in pathogen virulence, was not fully resolved. While we applaud the inclusion of the model shown in Fig 6F, the interdependence of these different stages of UmPR-1La action is not well supported by experimental data. In particular, we do not know whether cell wall association of UmPR-1La is important for virulence. While we know that that a mutation (FY*) in the CBM affect virulence (Fig 3E), we do not know whether cleavage of CAPE-La would repress host immunity, independently of the function of the CBM. The virulence of the FY* appears to be as repressed as that of a deletion, even though the FY* mutant version should still be able to release the CAPE-La peptide and thereby promote virulence. Thus, a more complete set of data defining the function of these different elements through which UmPR-1La appears to promote fungal virulence is needed to support the interdependence of the individual stages shown in Fig 6F, including abrogation of its cell wall association, mutation of the protease cleavage site (CNYx), and a truncation of the CAPE-La peptide.

In addition, the data supporting binding of the phenolic compound to UmPR-1La are still weak and subject to interpretation. Ligand binding is dose dependent and saturable. The authors do not provide strong support for either of these two criteria. They observe quenching of Trp fluorescence upon addition of phenolic compounds and interpret this as binding. These phenolic compounds could by themselves quench Trp fluorescence or induce partial denaturation of the protein rather than exhibiting a stoichiometric ligand-type of binding to the protein. In their revision, the authors have addressed the stability of the protein in the presence of the phenolic compounds but did not address the possibility that these phenolics induce partial denaturation or quench fluorescence. In addition, the authors do not show evidence that UmPR-1La binds Eugenol itself or whether this binding is dependent on the CBM. Overexpression of UmPR-1La confers resistance to Eugenol, and this resistance is affected by a mutation (FY*) in the CBM (Fig 3D). However, the FY* mutant grows better than WT in the presence of Ferulic acid (FA, Fig 4B), even though the FY* mutant does not "bind" FA in vitro (Fig 4F). Clearly, this is not consistent with the model proposed by the authors that UmPR-1La binds FA through its CBM to induce pseudohyphal growth and FA resistance. The same applies for the improved growth of the FY* mutant observed in presence of Coumaric acid (Fig S4A).

Reviewer #2 (Remarks to the Author):

My questions and concerns have been addressed sufficiently. The revised manuscript is significantly improved and major issues have been fixed / new experiments add clarifying and interesting information.

Reviewer #3 (Remarks to the Author):

The authors have addressed all of my concerns with the original manuscript.

Below are some minor edits to further improve the manuscript, some are just typos and some are to improve the figure legends further, as although greatly improved, they are not quite right in a couple of cases.

Minor corrections

Line 164: AF594-immunostained PR-1La – the figure legend also calls this AF594 however in the figure it is labelled AF595.

Figure 2a legend still needs improved. Authors only describe some of the constructs (Overexpression of WT constructs not listed) additionally, the list is in reverse order compared to the figure i.e. the last construct SG200_PRY1 is listed first in legend, please re-order. Please make the names in figure and legend the same SG200_PRY1 not PRY1

Figure 2d legend: Please indicate size of proteins, in the legend, to direct reader to correct bands especially important for Pr-1Lb and PRY1.

Figure S4d: Coomassie stain for loading control is missing.

Line 303: “Cathepsin B-like 302 3 (CatB3; GRMZM2G108849), which is expressed in the early stage and strongly induced from 4dpi and onwards (Fig. S5b).” please rephrase to include “of infection”

REVIEWER COMMENTS

We extend our gratitude to all the reviewers for their valuable constructive suggestions and comments, as well as their dedicated efforts to enhance the quality of our work.

Reviewer #1 (Remarks to the Author):

We appreciate the author's efforts to address our concerns and think that the revised version of this manuscript has been greatly improved. Our main concern, however, that the results do not provide a strong coherent picture of the function of UmPR-1La in pathogen virulence, was not fully resolved. While we applaud the inclusion of the model shown in Fig 6F, the interdependence of these different stages of UmPR-1La action is not well supported by experimental data.

In particular, we do not know whether cell wall association of UmPR-1La is important for virulence. Thank you for drawing our attention to this matter. We acknowledge that the current evidence falls short of directly demonstrating the essentiality of UmPR-1La's cell wall association for virulence. To establish this link definitively, we must first investigate how UmPR-1La localizes to the cell wall by identifying the specific cell wall component(s) it interacts with and the corresponding interacting motif. This endeavour would require a considerable amount of effort and resources.

Given the scope of this study, we find it more appropriate to reserve these experiments for our future research—Investigating how UmPR-1La induces hyphal-like structures and the cleavage mechanism. In the meantime, we think the evidence we have gathered so far (as elaborated below) substantiates our proposed working model, which emphasizes the critical role of UmPR-1La's cell wall association in its virulence.

- Given that PRY1 expression in SG200 cells cannot localize to the cell walls and protect cells, even though both the full-length proteins and the CAP domain alone can bind eugenols (Choudhary and Schneiter, 2012 PNAS; <https://doi.org/10.1073/pnas.1209086109>), we might anticipate that expressing a cell-wall localization-defective UmPR-1La protein with an intact CBM will similarly fail to protect cells. In contrast to ScPRY1, which localizes to the cell periphery of *S. cerevisiae* and shields them from eugenols (Fig. 2b), UmPR-1La primarily accumulates at the cell division sites of *U. maydis* sporidial cells (Fig. 2a). This localization pattern is insufficient to prevent eugenols from disrupting the cell membrane of *U. maydis*, unless they can lead to the formation of UmPR-1La-shielded hyphae for protection. This underscores the significance of localized UmPR-1La on the cell wall in inducing protective hyphal-like structures.
- The cell-wall localization of UmPR-1La doesn't ensure protection for sporidial cells. Despite their cell-wall localization, both the chimera and Sr10279 failed to induce hyphal-like structures and provide cell protection. This likely results from their inability to transmit crucial intracellular signals needed for shielded hyphae formation. The chimera's PRY1 CAP domain might not induce a conformational change in UmPR-1La's N-terminus, which may have an additional role in downstream partner interaction. Similarly, Sr10279 may struggle to interact with the necessary UmPR-1La partner for structure initiation. Our findings underscore a sequential process for initiating protective hyphae: proteins must first localize to *U. maydis* sporidial cell walls to detect signals and then transmit them intracellularly for hyphal-like structure induction. Failure in these steps won't restore $\Delta pr-1la$ virulence.
- By comparing the virulence phenotypes of complementation strains $\Delta pr-1la_PR-1La(FY^*)$ and $\Delta pr-1la_PR-1La(CAPE-Lb)$ (Fig 3e and 6d), we noted that the truncated PR-1La(CAPE-Lb) protein with the CAPE-Lb peptide partially restored $\Delta pr-1la$'s virulence phenotype. This happened because the truncated protein, carrying a defective peptide, retained the intact N-terminus and CBM, allowing it to localize to the cell wall and induce hyphal structures for protection, resulting in an intermediate virulence phenotype. Conversely, FY* proteins, unable to induce protective

hyphal-like structures, failed to complement $\Delta pr-1la$'s virulence. These findings underscore the crucial role of inducing hyphal structures for PR-1La's virulence function.

- To provide clarity for readers, we have incorporated this information into both the discussion and results sections. Presently, the mechanism through which UmPR-1La triggers the formation of hyphal structures for cell protection remains unclear. Further research is necessary to comprehend how the N-terminus and CAP domains contribute to signal transduction and, at a minimum, to pinpoint the involved receptor. This aspect is the primary focus of our ongoing investigation.

While we know that that a mutation (FY*) in the CBM affect virulence (Fig 3E), we do not know whether cleavage of CAPE-La would repress host immunity, independently of the function of the CBM. The virulence of the FY* appears to be as repressed as that of a deletion, even though the FY* mutant version should still be able to release the CAPE-La peptide and thereby promote virulence.

Thanks for raising this point. Please allow us to explain it.

- As mentioned above, hyphal formation plays a crucial role in the protective function of PR-1La, regardless of having an intact CNYx motif and CAPE-La peptides. Without hyphal formation, UmPR-1La mainly localizes at the site of cell division and cannot fully shield cells from plant toxic phenolics. Consequently, the $\Delta pr-1la_PR-1La(FY^*)$ complementation strain that unable to trigger hyphal formation compromised the $\Delta pr-1la$ virulence (fig.3E). This underscores the importance of inducing hyphal formation for survival within the plant, as this subsequently facilitates the release of the CAPE-La peptide, further suppressing plant immunity.
- As expected, the *in-vitro* cleavage assay showed that FY* mutant proteins could be cleaved by CatB3 (see western blot below). Based on our genetic data and biochemical assay, it is clear that the inability of FY* mutant proteins to restore virulence is due to their inability to sense phenolics and thereby fail to trigger the protective hyphae. As CatB3 selectively cleaves PR-1La proteins but not plant PR-1 and this cleavage mechanism is the focus of our next investigation, we have decided not to include this data in the revised manuscript. But, we provide it here for the reviewers.

Thus, a more complete set of data defining the function of these different elements through which UmPR-1La appears to promote fungal virulence is needed to support the interdependence of the individual stages shown in Fig 6F, including abrogation of its cell wall association, mutation of the protease cleavage site (CNYx), and a truncation of the CAPE-La peptide.

We wholeheartedly agree with the reviewer's comments. However, this requires further efforts for dissection, which is the focus of our next investigation. We believe that the findings in the discovery of the dual virulence functions and the underlying mechanism of UmPR-1La in sensing and adapting to the plant environment for survival, as well as countering plant defense to enhance the virulence of *Ustilago maydis*, not only provide important knowledge but also open a new research direction for the fields of effector biology and fungal-plant interactions.

- Regarding the truncation of CAPE-La, we have used a scrambled peptide of CAPE-La as a negative control to support the suppressive function of CAPE-La (Fig. 6e). Furthermore, we replaced UmPR-

1La's CAPE-La with CAPE-Lb to demonstrate the decrease in virulence (Fig. 6d). The results from these experiments had provided sufficient evidence to support our conclusion on the role of CAPE-La in suppressing plant immunity.

- As for the mutation on the CNYx motif, it has been shown that the tyrosine mutation in the plant PR-1's CNYD motif prevents the release of the AtCAPE9 peptide by AtXcp1 protease (<https://doi.org/10.21203/rs.3.rs-155784/v1>). However, in *Ustilago maydis*, a mutation on the Y residue (CNAD) abolishes the secretion of UmPR-1La. This finding is consistent with the work done by Tamara et al (<https://doi.org/10.1111/mpp.13187>) showing the PR1 trafficking dependent on the intact CNYD motif. Therefore, performing the *in-vitro* cleavage assay on this CNAD mutant protein is not biologically relevant, as it is not secreted.

In addition, the data supporting binding of the phenolic compound to UmPR-1La are still weak and subject to interpretation. Ligand binding is dose dependent and saturable. The authors do not provide strong support for either of these two criteria.

- We respectfully hold a different view from the reviewer concerning the binding strength of FA to PR-1La. It's also important to consider that the Y-axis scale is larger, potentially causing visual misinterpretation. The observed difference of approximately 250 a.u. (ranging from 1100 a.u. at 0 μ M FA to 850 a.u. at 10 μ M FA) does not necessarily imply weak binding. Notably, the similar fluorescence pattern shift was also reported in several protein-ligand binding assays using fluorescence spectrometry (<https://doi.org/10.1002/jsfa.11733>; <https://doi.org/10.1074/jbc.M111.301630>).
- The titration assay depicted in fig. S4b illustrated the dose-dependent response of PR-1La proteins to ferulic acids (FA), reaching saturation at approximately 10 μ M and exhibiting detectable free FA absorption at 410-430 nm. Conversely, the control PR-1La(FY*) proteins displayed no responsiveness to FA, showing no shifts in fluorescent intensity regardless of the added FA quantity. When considered alongside the information presented in fig. 4f, these findings indicate that the interaction between PR-1La and ferulic acids relies on the intact CBM in a manner that varies with the dose.

They observe quenching of Trp fluorescence upon addition of phenolic compounds and interpret this as binding. These phenolic compounds could by themselves quench Trp fluorescence or induce partial denaturation of the protein rather than exhibiting a stoichiometric ligand-type of binding to the protein. In their revision, the authors have addressed the stability of the protein in the presence of the phenolic compounds but did not address the possibility that these phenolics induce partial denaturation or quench fluorescence.

Thanks for asking these questions. Please allow us to explain our thought.

- If Trp fluorescence is quenched by phenolic compounds, we should also expect a similar fluorescent shift in the FY* proteins, but this was not observed in the FY* titration assay (fig. S4b).

- If the observed shift in the PR-1La protein was caused by partial denaturation, we would anticipate a decrease in the intensity of both PR-1La and the FY* mutant proteins, regardless of the type of phenolics. However, the intensity of FY* proteins did not decrease during incubation with ferulic acids. It is worth noting that the FY* proteins seemed less stable than the wild-type PR-1La proteins, as indicated by the detection of the truncated form (Fig. S4d). Furthermore, we also noticed an increase in the intrinsic fluorescence intensity of PR-1La when incubated with coniferyl alcohol (Fig. S4c). Based on these results and the induced hyphal-like structure phenotype in Fig 3d, this allows us to suggest that the PR-1La is able to bind phenolics.

In addition, the authors do not show evidence that UmPR-1La binds Eugenol itself or whether this binding is dependent on the CBM.

Due to the lipophilic property of eugenols, it exhibits strong fluorescence, making it unsuitable for demonstrating the binding of PR-1La to phenolics using fluorescence spectrometry or isothermal titration calorimetry (ITC). Since cells expressing UmPR-1La responded to both eugenols and ferulic acids by forming similar hyphal-like structures, we used ferulic acids to demonstrate the phenolic binding of PR-1La.

Overexpression of UmPR-1La confers resistance to Eugenol, and this resistance is affected by a mutation (FY*) in the CBM (Fig 3D). However, the FY* mutant grows better than WT in the presence of Ferulic acid (FA, Fig 4B), even though the FY* mutant does not “bind” FA in vitro (Fig 4F). Clearly, this is not consistent with the model proposed by the authors that UmPR-1La binds FA through its CBM to induce pseudohyphal growth and FA resistance. The same applies for the improved growth of the FY* mutant observed in presence of Coumaric acid (Fig S4A).

The survival of FY*-expressing cells in the presence of phenolics depends on the toxicity level of the tested compounds. Cells cannot survive in the presence of eugenols due to the disruption of the cell membrane caused by eugenols, as documented in the reference (DOI: 10.3109/13693786.2012.742966). Therefore, the cell membrane needs to be protected thru triggering UmPR-1La-shielded hyphae to block eugenols. The survival of cells in the presence of ferulic acids/coumaric acids, on the other hand, is observed to be higher, indicating their lower toxicity to *U. maydis* cells under the tested conditions. Despite the inability of FY* proteins to induce hyphae structures, FY*-expressing cells still manage to survive and multiply in the presence of FA-containing medium. This observation supports our proposal that UmPR-1La can sense various plant phenolics, regardless of their toxicity level, and trigger UmPR-1La-shielded hyphal branching to confer protection within the plant.

Reviewer #2 (Remarks to the Author):

My questions and concerns have been addressed sufficiently. The revised manuscript is significantly improved and major issues have been fixed / new experiments add clarifying and interesting information.

Thank you.

Reviewer #3 (Remarks to the Author):

The authors have addressed all of my concerns with the original manuscript.

Below are some minor edits to further improve the manuscript, some are just typos and some are to improve the figure legends further, as although greatly improved, they are not quite right in a couple of

cases.

Minor corrections

Line 164: AF594-immunostained PR-1La – the figure legend also calls this AF594 however in the figure it is labelled AF595.

Thanks for pointing it out. It has been corrected.

Figure 2a legend still needs improved. Authors only describe some of the constructs (Overexpression of WT constructs not listed) additionally, the list is in reverse order compared to the figure i.e. the last construct SG200_PRY1 is listed first in legend, please re-order. Please make the names in figure and legend the same SG200_PRY1 not PRY1.

We have made the changes accordingly.

Figure 2d legend: Please indicate size of proteins, in the legend, to direct reader to correct bands especially important for Pr-1Lb and PRY1.

Thank you. We have added the expected size of full-length proteins in the legend of fig 2d.

Figure S4d: Coomassie stain for loading control is missing. We have now included the loading controls in fig. S4d.

Line 303: “Cathepsin B-like 302 3 (CatB3; GRMZM2G108849), which is expressed in the early stage and strongly induced from 4dpi and onwards (Fig. S5b).” please rephrase to include “of infection”

The sentence has been rephrased as suggested.

REVIEWERS' COMMENTS

Reviewer #1 (Remarks to the Author):

Thank you for having addressed our remaining concerns.